# Past and recent anthropogenic pressures drive rapid changes in riverine fish communities

**Alain Danet** [1,4] ✉, **Xingli Giam** [2], **Julian D. Olden** [3] **& Lise Comte** [1]

Understanding how and why local communities change is a pressing task for conservation, especially in freshwater systems. It remains challenging because of the complexity of biodiversity changes, driven by the spatio-temporal heterogeneity of human pressures. Using a compilation of riverine fish community time series (93% between 1993 and 2019) across the Palaearctic, Nearctic and Australasia realms, we assessed how past and recent anthropogenic pressures drive community changes across both space and time. We found evidence of rapid changes in community composition of 30% per decade characterized by important changes in the dominant species, together with a 13% increase in total abundance per decade and a 7% increase in species richness per decade. The spatial heterogeneity in these trends could be traced back to the strength and timing of anthropogenic pressures and was mainly mediated by non-native species introductions. Specifically, we demonstrate that the negative effects of anthropogenic pressures on species richness and total abundance were compensated over time by the establishment of non-native species, a pattern consistent with previously reported biotic homogenization at the global scale. Overall, our study suggests that accounting for the complexity of community changes and its drivers is a crucial step to reach global conservation goals.

Biological communities are undergoing dramatic reassembly in response to an array of ever-growing human impacts[1]. Changes in species composition and not necessarily systematic reductions in local-scale species richness are becoming increasingly recognized[2,3], often resulting in ecosystem consequences manifested across large spatial scales[4,5]. Repeated calls have been made for greater scientific clarity regarding how heterogeneous rates of species losses and gains across space may shift community structure over time[6]. Advancing this knowledge is particularly relevant for freshwater ecosystems, where vertebrate populations are declining substantially faster than those in terrestrial or marine systems[7].

Land use conversion is a persistent and pervasive threat to freshwater ecosystems[8] with striking repercussions for freshwater fish biodiversity[9,10]. Dense urban and cultivated areas are often associated with reduced species richness and abundance[11,12], and shifts in local community composition towards more tolerant and ubiquitous species that can cope with degraded conditions[13,14]. Non-native species can also play a disproportionate role in the reassembly of communities over time[5,15–17], and have dramatic effects on native species when they become invasive[18], including the widespread homogenization of faunas[19]. Hubs of human activities such as human settlements, transport and trade are also responsible for major habitat alterations and

[1]School of Biological Sciences, Illinois State University, Normal, IL, USA. [2]Department of Ecology and Evolutionary Biology, The University of Tennessee, Knoxville, TN, USA. [3]School of Aquatic and Fishery Sciences, University of Washington, Seattle, WA, USA. [4]Present address: School of Biosciences, University of Sheffield, Sheffield, UK. ✉e-mail: a.h.danet@sheffield.ac.uk

increased accessibility, resulting in more frequent non-native intro-duction events and opportunities for spread[20–22]. Human activities may therefore have opposing effects on local diversity by decreasing the number and abundance of native species, while concurrently pro-moting the establishment and spread of non-native species that can increase community total abundance and species richness[4,6]. Under-standing community changes therefore requires going beyond analyses of changes in the number of species or individuals by considering concomitant changes in species identity[2,3,23,24].

Temporal changes in community composition are influenced by past anthropogenic pressures that can generate transient eco-logical dynamics and long-lasting biotic 'legacies'[25]. Given the high spatio-temporal heterogeneity of anthropogenic pressures[26], ignor-ing the long-term antecedent effects of historical pressures and their recent changes can greatly impede our understanding of the drivers of community change, such as what has been demonstrated for the effects of invasive species[27]. Additionally, habitat structure and connectivity can enhance or dampen community responses to anthropogenic pres-sures by mediating dispersal among habitats[28]. Accounting for past and recent anthropogenic pressures as well as spatial distribution of habi-tats may therefore improve our understanding of community changes.

This study investigates the spatio-temporal changes of riverine fish communities in response to human pressures from local to conti-nental extents. To do so, we leveraged a compilation of 4,476 riverine fish community time series[29] that had been repeatedly sampled from 1957 to 2019 using variable durations and frequencies (93% of the samplings between 1993 and 2019 with a minimum of 5 years of sam-pling; Extended Data Fig. 1), mainly using electrofishing (98% of the samplings). The sites are located in various river basins, mainly across the Palaearctic, Nearctic and Australasia realms (99.9% of the sites). We used Bayesian hierarchical models to assess temporal changes in total abundance, species richness and community composition across local communities, including in the share of non-native species. We next characterized the typology of community temporal trends by examining the covariations among different community metrics, and identifying trajectories of community change across spatial scales. We finally assessed how fish community changes could be traced back to the spatio-temporal changes in anthropogenic pressures and longi-tudinal stream position. Anthropogenic pressures were quantified with the human footprint index, which includes an array of pressures such as population density, land use and human-built infrastructure[30] (Methods), and has been previously related to species extinction and invasion risks[31,32]. Outcomes of this study further our understanding of the complexity of local community changes by addressing the effects of global change and advancing new knowledge that can inform actions seeking to curb the current freshwater biodiversity crisis.

## Results

### Community temporal trends
Riverine fish communities have demonstrated remarkable change over recent decades (Fig. 1; range first survey year: 1957–2010, median = 1997; time span: 10–60 years, median = 17; see Extended Data Fig. 1 for more details on the time series). We estimated temporal trends with a hierarchical Bayesian modelling approach that accounts for spatial variation at both hydrographic river basin and site levels (that is, by including random terms on the intercept and temporal trends; see Methods for a detailed description of the models), finding that com-munities have increased in both total abundance and species richness, but decreased in the proportional abundance of non-native species. We further found that the estimated temporal trends were not influenced by the characteristics of the time series, such as the temporal span, survey completeness and starting year (Extended Data Fig. 2). From this model, we considered weak, medium and strong evidence for an effect when its credible interval at respectively 80, 90 and 95% did not overlap zero[33,34]. We found strong evidence for an average increase in total

community abundance (average credible interval (CI) 95%: 13.2% (2.9%, 23.8%) per decade; Fig. 1a) and in species richness (CI 95%: 6.9% (3.9%, 9.9%) per decade; Fig. 1b) over time. By contrast, we found an average decline in the proportion of non-native species abundance (moderate evidence, CI 90%: −0.0047 (−0.0091, −0.0004) per decade; Fig. 1c), and no evidence for a temporal trend in the proportion of non-native species richness (CI 80%: 0.001 (−0.001, 0.004) per decade; Fig. 1d).

Changes in abundance and species richness were accompanied by rapid compositional reorganization, with an average decline in com-munity similarity of about 30% per decade when considering either species abundances (Simpson dissimilarity, CI 95%: 0.33 (0.31, 0.34) per decade, hereafter 'temporal dissimilarity'; Fig. 1e) or occurrences (Jaccard dissimilarity, CI 95%: 0.31 (0.30, 0.33) per decade; Extended Data Fig. 3a). The consistency in the Simpson and Jaccard dissimilarity metrics indicated that changes in temporal dissimilarity resulted from changes in the identity of the dominant rather than of the rare spe-cies. The partitioning of the Jaccard dissimilarity index into turnover, describing composition changes arising from species replacement, and nestedness, describing changes arising from species gains or losses from a common species pool, further showed a comparable increase over time (CI 95%: 0.17 (0.16, 0.18) and 0.16 (0.15, 0.17) per decade respectively; Fig. 1f and Extended Data Fig. 3b). This suggests that changes in community composition were driven by species replace-ment in the community, in addition to species losses or gains.

Beyond overall temporal trends, considerable spatial heterogenei-ty exists across sites, as illustrated by the spread of the histograms in Fig. 1. This heterogeneity is also apparent within the same river basin (Supplementary Software 1). For example, on average the Thames basin shows the same spatial patterns as at the global scale, but a variety of temporal trends were observed across the 139 sites within the basin, including decrease in species richness (18 sites), decrease in abun-dance (20) and high turnover (17). The (random) slope of time in our hierarchical models varied much more (that is, up to more than twice as much) across sites within river basins than across different basins for all community metrics (Extended Data Table 1). This suggested that relatively finer-scale environmental variation within river basins has a greater effect on community changes than larger-scale environmental or biogeographical variation across river basins.

### Typology of community temporal trends
We further assessed covariations among the temporal trends of differ-ent community metrics to identify potential 'types' of community tem-poral trajectory, using the temporal trends at the site level estimated from the hierarchical Bayesian model. There was a moderate level of association among the different community trajectory metrics; the first two axes of the principal component analysis (PCA) explained 69% of the total variability among fish communities (Fig. 2a–b). Temporal trends in community composition (that is, temporal dissimilarity and turnover) were positively associated with each other, as were temporal trends in total abundance and species richness; however, these two sets of trajectories appeared largely independent of each other (Fig. 2a). Using a k-mean trimmed clustering method on the temporal trends in the community metrics at the site level (Methods), we further detected six distinct types of community trajectory (Fig. 2c; non-assigned sites are displayed in Extended Data Fig. 4). The largest cluster was character-ized by moderate changes along all biodiversity dimensions: medium temporal increases in total abundance and species richness, tempo-ral dissimilarity, and turnover ('medium change'; 42% of the sites). The second cluster was associated with communities showing strong turnover but moderate increases in total abundance, species richness and temporal dissimilarity ('high turnover'; 16% of the sites). The third, fourth and fifth clusters were characterized by temporal community changes along a single dimension: a strong increase in species richness ('increase in species richness'; 13% of the sites), a strong decline in total abundance ('decrease in total abundance'; 12% of the sites) or a strong

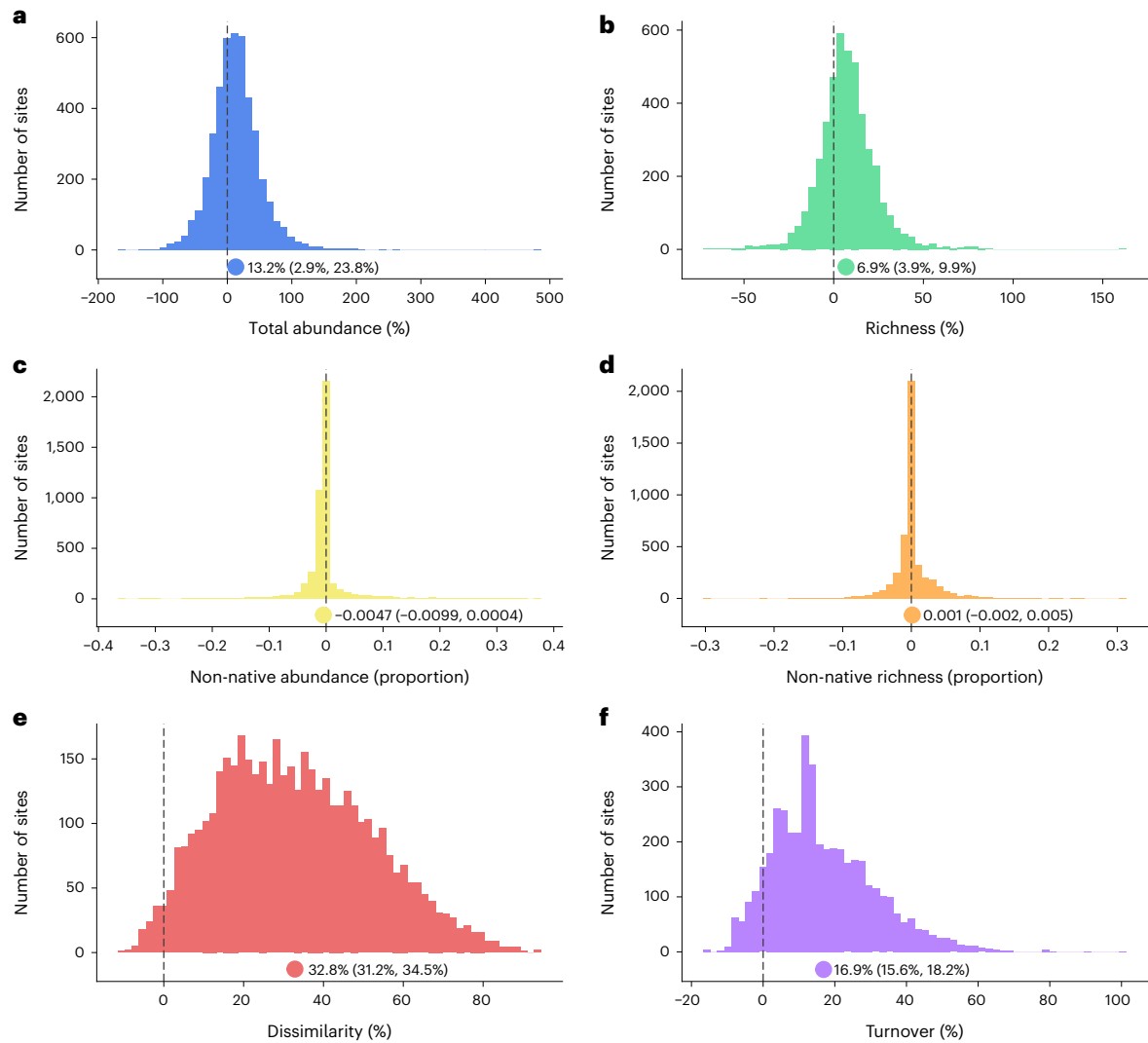

**Fig. 1 | Distribution of community temporal trends per decade across sites. a–f**, Total abundance (**a**), species richness (**b**), proportion of non-native total abundance (**c**), proportion of non-native species richness (**d**), Simpson temporal dissimilarity (**e**) and Jaccard turnover (**f**). Temporal trends per decade were estimated from a hierarchical Bayesian model including time as sole fixed predictor and using a random slope to estimate temporal trends at each site (Methods). The histograms show the best linear unbiased predictor estimated at each site and the dots below the histograms represent the average posterior distribution with labels depicting the Bayesian CI at 95%. The dashed lines denote no temporal trend. *N* = 46,932 sampling events across 4,476 sites.

decline in species richness ('decrease in species richness'; 9% of the sites), respectively. The last and smallest cluster was associated with communities that remained relatively stable over time ('low change'; 7% of the sites). The relative frequency of the different community trajectories was broadly similar across the three main biogeographic realms (Fig. 2d).

### Drivers of community temporal trends

We detected complex synergies between the legacy of past anthropogenic pressures and the effects of recent anthropogenic pressures on community temporal trends, by considering additional predictors associated with the human footprint index and longitudinal stream position (Fig. 3a; see model predictions in Extended Data Fig. 5). In addition, we found that these additional predictors were not related to the characteristics of the time series (Extended Data Fig. 6). Specifically, we found strong evidence that a higher degree of past anthropogenic pressures (that is, human footprint index of 1993 corresponding to the beginning of the time series) was associated with faster increases in total abundance and species richness (respective

CI 95%: 0.02 (0.01, 0.04) in blue and 0.03 (0.01, 0.05) in green; Fig. 3a). We also uncovered evidence for an interaction with the longitudinal stream position (that is, represented by a synthetic PCA axis based on several hydromorphological characteristics where high values are associated with more downstream areas; Extended Data Fig. 7), such as the legacy effects of past anthropogenic pressures on total abundance (strong evidence) and species richness (weak evidence) were buffered in more downstream areas (respective CI 95%: $\beta'$ =−0.021 (−0.031, −0.011) and CI 80%: −0.0108 (−0.0191, −0.0025); Fig. 3a). Similar results were obtained using raw or coverage-based species richness (Methods and Extended Data Fig. 8).

Past anthropogenic pressures were also associated with changes in community composition. We found evidence (albeit weak) that a higher degree of past anthropogenic pressures was associated with an increase in the proportion of non-native richness over time (CI 80%: 0.02 (0.00, 0.04) in orange; Fig. 3a), and that this effect was enhanced in more downstream areas (CI 90%: 0.023 (0.003, 0.043); Fig. 3a). Although we found no overall associations between past anthropogenic pressures and temporal trends in non-native species abundance (CI 80%: 0.00

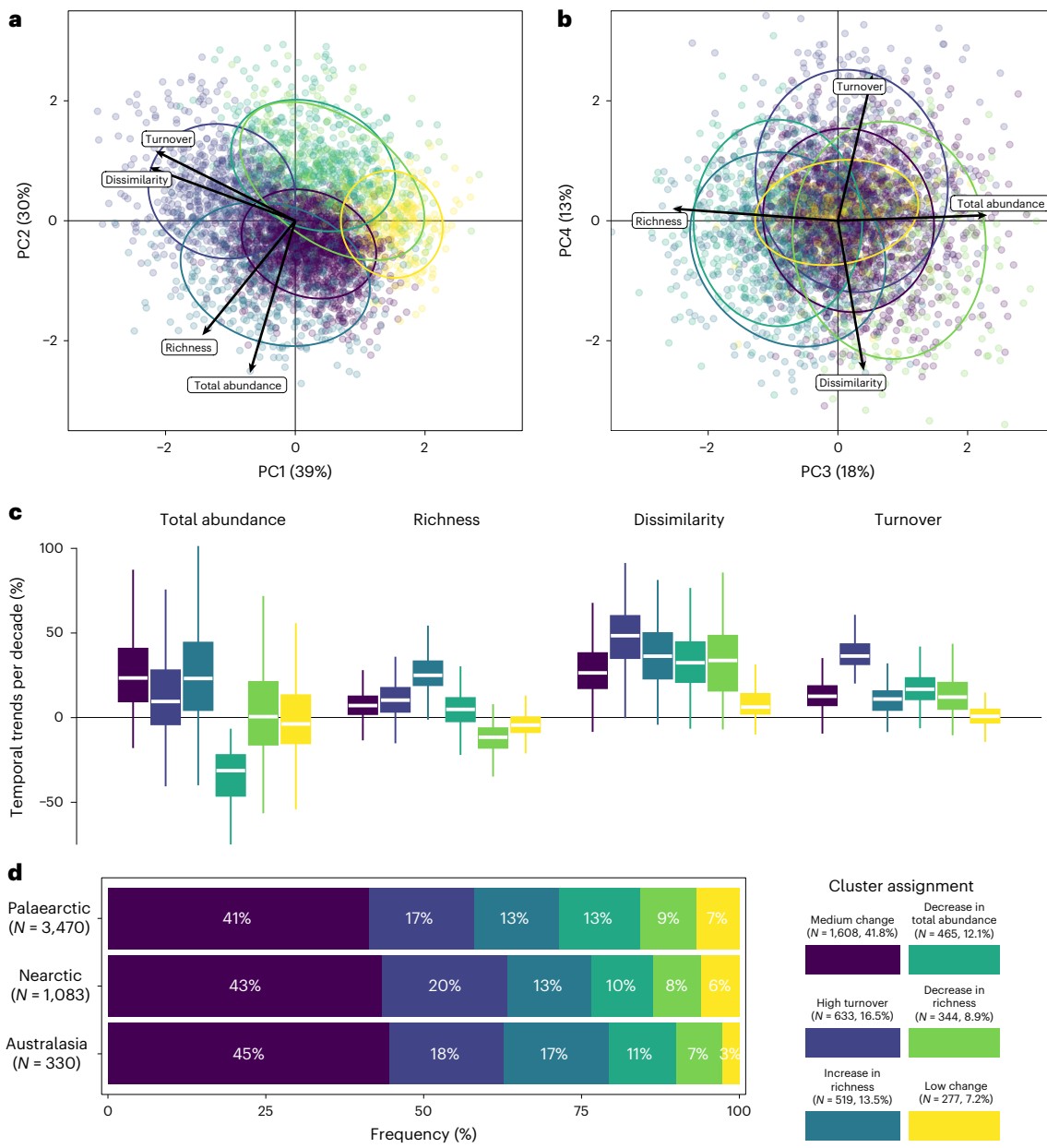

**Fig. 2 | Covariation among the community temporal trends and characterization of community trajectories. a,b**, PCA biplot of the community temporal trends and their cluster assignment where the sites are coloured according to their cluster assignment (**a**, first and second PCA axes; **b**, third and fourth PCA axes). **c**, Boxplots displaying the distribution of the temporal trends by cluster. The centre of the box depicts the median while the bounds depict the 25% and 75% percentiles. The whiskers depict the extreme values within 1.5× interquartile range beyond the bounds of the box. **d**, Cluster frequencies across the three main biogeographic realms. The ellipses in **a** and **b** display the 95% intervals around the clusters assuming a Student's *t* distribution. The clusters were named according to the most noticeable characteristic of changes across all the biodiversity metrics (**c**). In particular, 'medium change' cluster was associated with sites presenting moderate changes along all the biodiversity metrics considered. Sites not assigned to a cluster because of affiliation uncertainty (*N* = 641, 14%) are displayed in Extended Data Fig. 4. *N* = 46,932 sampling events across 4,476 sites.

(−0.02, 0.03) in yellow; Fig. 3a), we found moderate evidence that a higher degree of past anthropogenic pressures resulted in a higher rate of increase in non-native species abundance in the most downstream areas (CI 90%: 0.033 (0.003, 0.064); Fig. 3a). We also found that a higher degree of past anthropogenic pressures was associated with faster rates of increases in temporal dissimilarity and turnover (respective CI 95%: 0.23 (0.20, 0.27) in red and 0.25 (0.21, 0.29) in purple; Fig. 3a), irrespective of the longitudinal stream position (respective CI 80%: −0.017 (−0.036, 0.002) and −0.016 (−0.037, 0.004); Fig. 3a). This result was consistent between Jaccard and Simpson-based dissimilarity (Extended Data Fig. 8), but the effects of past anthropogenic

pressures were attenuated in most downstream areas for Jaccard's dissimilarity, indicating that those changes involved more dominant species (Extended Data Fig. 8).

Recent increases in anthropogenic pressures (that is, ratio of the human footprint index between 2009 and 1993) were found to have a context-specific effect on total abundance and species richness, as well as to hasten community reorganization through increases in the share of non-native species and faster rates of temporal dissimilarity and turnover (Fig. 3a; see model predictions in Extended Data Fig. 5). More specifically, we found strong evidence of an antagonistic effect between past and recent anthropogenic pressures on total abundance

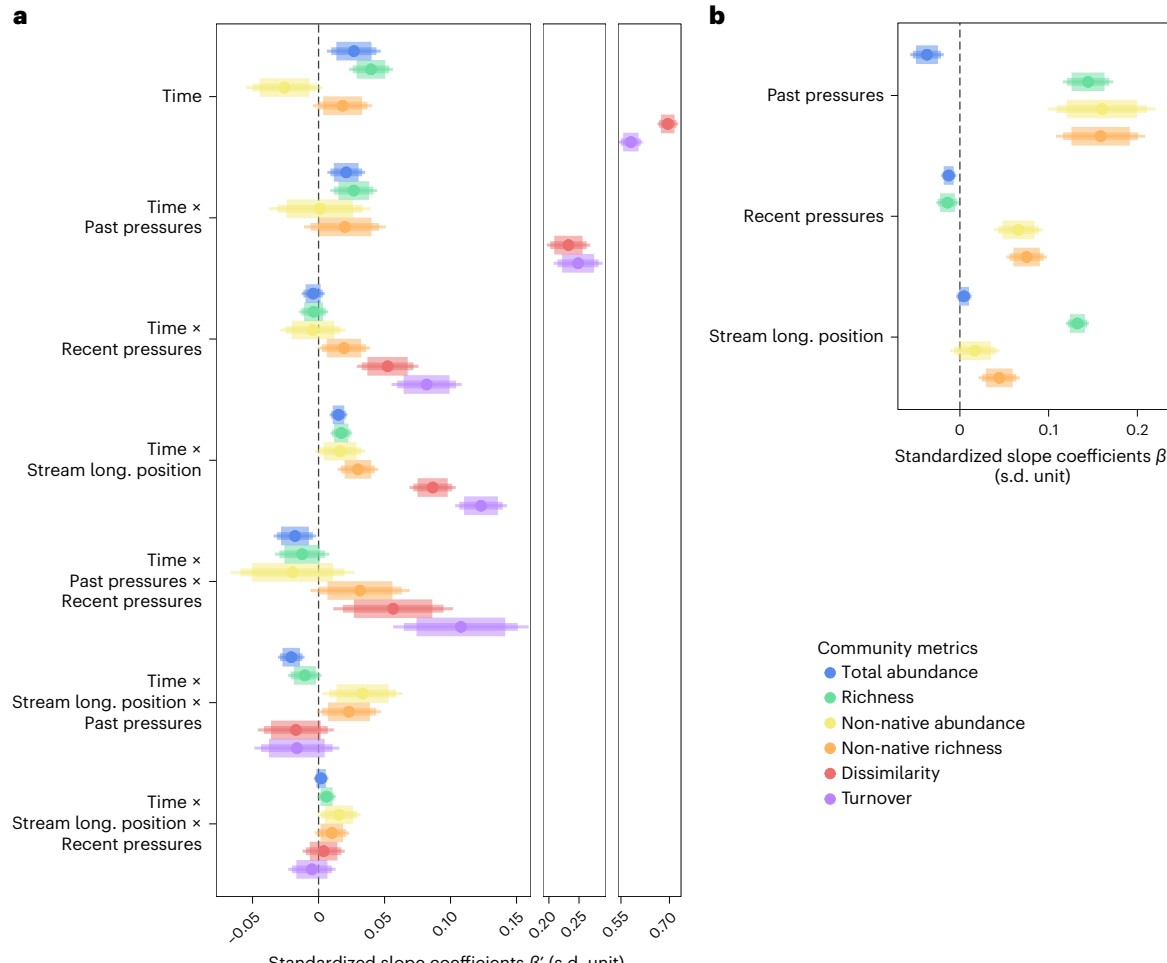

**Fig. 3 | Drivers of temporal change and spatial variation in fish community metrics. a,b,** Effects of anthropogenic pressures and longitudinal (long.) stream position on temporal changes (**a**) and spatial variation (**b**) in fish community metrics. Community metrics include total abundance, species richness, proportional abundance and richness of non-native species, Simpson temporal dissimilarity and Jaccard turnover. Points depict the average posterior distributions. Large, medium and thin bars depict the Bayesian CI at 80, 90 and 95%, respectively. Please note the broken abscissa scale in **a**. $N$ = 46,932 sampling events across 4,476 sites.

(CI 95%: −0.018 (−0.034, −0.002); Fig. 3a), such as the rate of increase observed across the historically most degraded sites was lower when these sites experienced a recent increase in anthropogenic pressures, although recent anthropogenic pressures per se had no effect on the temporal trends in total abundance (CI 80%: −0.004 (−0.010, 0.002); Fig. 3a). Similarly, recent changes in anthropogenic pressures did not have an overall effect on the temporal trends in species richness (CI 80%: −0.005 (−0.012, 0.003); Fig. 3a), but we found moderate evidence for a positive interaction with the longitudinal stream position (CI 90%: 0.0063 (0.0004, 0.0121); Fig. 3a), indicating that recent increases in anthropogenic pressures were associated with faster increases in species richness over time in more downstream areas.

Recent increases in anthropogenic pressures were also associated with more rapid increases in the proportion of non-native species abundance in more downstream areas (CI 90%: 0.015 (0.002, 0.029); Fig. 3a), although recent changes in anthropogenic pressures had no overall effect (CI 80%: −0.004 (−0.010, 0.002); Fig. 3a). When considering the share of non-native species richness, we found moderate evidence that a recent increase in anthropogenic pressures was associated with an increase in the proportion of non-native species (CI 90%: 0.02 (0.00, 0.04); Fig. 3a). This effect was particularly pronounced in the historically most degraded sites and in the most downstream areas (respective CI 90%: 0.032 (0.000, 0.063) and 0.010 (0.001, 0.019);

Fig. 3a). In addition, we found strong evidence that a recent increase in anthropogenic pressures resulted in stronger temporal dissimilarity and turnover (respective CI 95%: 0.05 (0.03, 0.08) and 0.08 (0.06, 0.11); Fig. 3a). These effects were hastened in the most historically degraded sites (respective CI 95%: 0.06 (0.01, 0.10) and 0.11 (0.06, 0.16); Fig. 3a), but not affected by the longitudinal stream position (respective CI 80%: 0.00 (−0.01, 0.01) and −0.01 (−0.02, 0.01)). By contrast, our results indicated that an increase in recent anthropogenic pressures in the historically most degraded sites was associated with slower rates of increase in total abundance and species richness, but faster rates of increase in non-native richness, temporal dissimilarity and turnover (see model predictions in Extended Data Fig. 5).

### Drivers of community variation across space

Spatial variation in community structure was strongly associated with past and recent anthropogenic pressures and with longitudinal stream position (that is, single model effects independent of time; Fig. 3b). Using baseline model prediction (that is, at $t$ = 0; Extended Data Fig. 5 and Methods), we found that a higher degree of past anthropogenic pressures was associated with lower total abundance (strong evidence; Fig. 3b, blue), with the most 'degraded' sites (that is, with a human footprint index = 45.6) displaying a total abundance 30% lower than the most 'intact' sites (that is, with a human footprint index = 2.5). By contrast,

**Table 1 | Marginal and conditional $R^2$ of the hierarchical Bayesian model for each community metric**

| Response variable | Marginal $R^2$ | Conditional $R^2$ |
|---|---|---|
| Total abundance | 0.03 (0.01, 0.07) | 0.73 (0.72, 0.74) |
| Richness | 0.02 (0.01, 0.05) | 0.81 (0.80, 0.81) |
| Non-native richness | 0.03 (0.01, 0.06) | 0.75 (0.75, 0.76) |
| Non-native abundance | 0.03 (0.01, 0.07) | 0.76 (0.76, 0.77) |
| Dissimilarity | 0.07 (0.02, 0.15) | 0.33 (0.29, 0.38) |
| Turnover | 0.07 (0.01, 0.16) | 0.14 (0.08, 0.22) |

The models included several predictors (fixed effects illustrated in Fig. 3) and accounted for spatial variations at the hydrographic river basin and site levels (random effects). Marginal $R^2$ accounts for fixed effects and conditional $R^2$ accounts for both fixed and random effects. Mean $R^2$ (95% CI): CI computed using the highest posterior density method.

a higher degree of past anthropogenic pressures was associated with higher species richness (strong evidence; Fig. 3b, green), with the most degraded sites displaying 64% more species than the most intact sites.

Recent increases in anthropogenic pressures were strongly associated with lower total abundance and species richness (Fig. 3b). More specifically, sites that experienced a twofold increase in recent anthropogenic pressures had 16% lower total abundance and 7% lower species richness than sites that had not undergone such pressures. Longitudinal stream position was strongly associated with species richness—the most downstream sites displayed three times more species than the most upstream sites (Fig. 3b).

Anthropogenic pressures and longitudinal stream position were associated with spatial variation in the proportion of non-native fish species. Sites that had experienced a higher degree of past and recent anthropogenic pressures had a higher proportion of non-native individuals and species (strong evidence; Fig. 3b, yellow and orange). The proportion of non-native individuals and species was three times (9% versus 3%) and two times (10% versus 4%) higher in the most degraded sites than the most intact sites. Further, a twofold increase in recent anthropogenic pressures was associated with an increase in the proportion of non-native individuals and species by 69% and 63%, respectively. The most downstream sites had 33% higher proportion of non-native abundance and 78% higher proportion of non-native richness than the most upstream sites (Fig. 3b). Noteworthy, a larger share of the variance in the community metrics was explained by site and basin identity rather than by the fixed effects alone ($R^2$ conditional varying from 0.15 for turnover to 0.80 for species richness versus $R^2$ marginal varying from 0.02 for species richness to 0.07 for community turnover; Table 1). This indicates that context dependencies are well captured by our hierarchical models but suggests that integrating fine-scale local drivers may further improve our ability to predict local community changes.

## Discussion

Recent decades have witnessed substantial shifts in riverine fish communities characterized by marked increases in species richness and total abundance over time, accompanied by a strong pattern of species replacement. We found that fish species richness has increased at a rate of ~7% per decade, although no net change in species richness had been previously reported in terrestrial and in marine systems[2,3,35]. We also found an overall increase in total fish abundance of ~13% per decade, which is in line with the increase of 11% per decade reported for freshwater insects[33]. This is also consistent with several regional assessments of freshwater population trends in the Palaearctic, such as the reported increase in freshwater insect occupancy documented in the UK or the increase in freshwater animal Living Planet Index in the Netherlands since the 1990s[36,37]. However, this finding contrasts with dramatic Living Planet Index declines reported at the global scale for freshwater species and particularly fish megafauna, as well as with

other regional assessments of fish assemblages[8,38]. We further found a faster average temporal trend in Jaccard dissimilarity (31% versus 10% per decade) but a slower average turnover (17% versus 28% per decade) than previously reported across a diversity of marine, freshwater and terrestrial assemblages[2,3], indicating that riverine fish communities experienced both important richness and compositional changes in recent decades.

These recent biotic changes are linked to complex spatio-temporal processes involving past and recent human impacts on the environment and their interaction with stream network position. Higher past anthropogenic pressures were associated with faster rates of species richness and total abundance increases over time, suggesting a recovery from the legacy of past disturbances. Previous studies suggested that the adoption of numerous legislations targeting improvements in water quality in the European Union and the United States since the 1970s, as well as a decrease in the negative effects of agriculture, could be partly responsible for those increases despite the surrounding habitat changes[33,39,40]. The fact that most of the study sites (92%) were already highly degraded at the beginning of the study period, that is, they had a human footprint index > 4 in 1993[41], could lend support to the recovery hypothesis.

However, a higher degree of past anthropogenic pressures was also associated with a higher share of non-native species; this effect being stronger in downstream sites. This indicates that the introduction and establishment of non-native species contributed most substantively to the fish community changes through time in the sites that suffered the greatest past (pre-1993) degradation, and particularly the most downstream ones. The increase in local species richness over time in degraded rivers could thus result from introduction of non-native species from ongoing spatial homogenization[17], a pattern well-supported by metacommunity models[42] and already documented across river basins in the Nearctic and Palaearctic realms[17]. This is in line with findings that higher densities of human population, urban areas and roads—all included in the human footprint index—can promote non-native species richness by increasing the number and frequency of introduction events[15,16,31]. Anthropogenic pressures can also alter the instream habitat to be more conducive for non-native species that are often ubiquitous and habitat generalists[13,14], giving them a competitive advantage over native species that are less suited to the new conditions[16,43–45]. A higher degree of past anthropogenic pressures was also associated with faster rates of species replacement and shift in species dominance over time. This suggests that the legacy effects of past habitat degradation are characterized by shifts towards species that are better adapted to degraded environments[9], to which non-native species contribute disproportionately[2,3].

This study uncovered important interaction effects between past and recent human pressures in driving the rate of change in several community metrics, highlighting the importance of considering both the degree and timing of anthropogenic pressures. For example, as discussed above, communities that experienced greater past degradation had actually experienced an increase in richness in recent years. But an increase in recent human pressures at these sites was associated with an increase of non-native species and lower species richness. This suggests that any recovery of the native fish communities in previously degraded sites would be severely compromised if human impacts were allowed to continue. Conversely, these findings clearly illustrate opportunities to reduce human impacts in previously degraded habitats to benefit freshwater biodiversity[7]. In turn, the fact that non-native species were more abundant in both historically or recently degraded sites, but that no direct association was uncovered in terms of non-native temporal trends, can be explained, at least in part, by commonly reported time lags between the first recorded introductions and the establishment of self-sustaining populations, which includes time for biological acceptance and local adaptation[16]. Our results demonstrate that recent habitat degradation can result in

simultaneous negative and positive effects on native and non-native species, respectively[6,46], and highlights the conservation challenges associated with the identification and management of biodiversity changes in the context of transient community dynamics[25].

Longitudinal position along the river network was found to mediate temporal biodiversity trends, with the most downstream sites experiencing faster rates of community change over time. This finding may be explained by the higher connectivity of larger rivers with other tributaries, which in turn gives more opportunity for local colonization[28] and the establishment of metacommunity dynamics[47]. As such, it is not entirely surprising that community changes were found to be more heterogeneous at the local scale than at the basin or realm scales, and that the spatial structure of the model explained much more of the variance in the community metrics than the fixed effects. This probably reflects the characteristics of riverine habitats, and especially their dendritic structure and isolation within drainage basins, which determine environmental filtering and dispersal opportunities[48,49]. In addition, we focused chiefly on community reorganization arising from land use pressures, therefore disregarding the potential interactions with other global drivers of change such as climate change and more localized threats such as water withdrawals[40,50].

Our results further confirm that temporal changes in species composition can be decoupled from changes in species richness in freshwater systems, similarly to what has been observed in mostly marine and terrestrial assemblages[2,3]. Various community trajectories can be linked to a complex mosaic of ecological drivers such as the degree and timing of anthropogenic pressures and position of the sites within watersheds. It follows that the similar frequencies in community trajectories detected across realms, together with the restricted number of sites displaying a low degree of change, probably reflects the spatially and temporally heterogeneous patterns in human pressures. We recognize that recent increases in anthropogenic pressures are most prevalent in tropical biodiversity hotspots[30,51], while our study has a spatial coverage limited to historically industrialized countries and in mostly temperate biomes. Noticeably, anthropogenic pressures decreased by 4% on average (based on differences in the human footprint index between 1993 and 2009) across the sites included in our study, while it increased by 25% across the rivers globally[52]. Consequently, species richness and abundance increases as well as the decoupling of compositional changes from richness changes may not be universal phenomena, and more heterogeneous patterns of biotic change may manifest at the global scale once we consider tropical fish communities. Our study remains essentially correlative. Although aggregated anthropogenic threat indices have been shown to be useful to estimate the ecological integrity of freshwater ecosystems[53], they do not replace the use of more targeted threat indices related to water quality and ecosystem functioning[54,55]. Nonetheless, our analyses were conducted on the best available temporal riverine fish dataset at this time, and they provide evidence linking the multidimensionality of community changes to the interplay between past and recent environmental challenges as well as habitat context. Our study also offers a framework for future research that merges multiple scales of both time and space[4,6], which could be leveraged as more tropical data are collected and become available.

In conclusion, our study uncovered complex but consistent effects of past and recent changes in anthropogenic pressures and stream network position on riverine fish communities. We showed that the timing of anthropogenic pressures matters, because past and recent pressures can have contrasting and interactive effects on community trends, partly mediated by non-native species. Our study further shows that considering multiple biodiversity facets can shed light on the complex mechanisms by which communities change over time. Looking forward, we emphasize the increasing need to investigate biotic changes across spatial scales to better reconcile reported local gains and global declines in biodiversity[4,6,56]. The increasing availability of community time series and environmental data across large areas is invaluable for understanding how human pressures impact biodiversity across taxa and ecosystems, and for implementing conservation policies to mitigate these impacts.

## Methods

### Fish community time series[29]

We used the RivFishTIME database[29], a compilation of more than 12,000 time series containing species abundances of riverine fish communities, which we completed with time series from Canada and the United States (Supplementary Table 2). The final database mainly covered western and northern Europe, northern America, and southeastern Australia. We selected time series having at least 5 years of data over a 10-year period as well as a consistent sampling protocol and abundance unit. As several sites had been sampled using different sampling methods (for example, electrofishing, seining; Supplementary Table 2), we selected for each site only the sampling events that were performed using the most frequent sampling method. To minimize the influence of seasonal variation in community composition (for example, due to spawning or migration), we further only selected sampling events that were performed within 1.5 months of the most frequently sampled month (that is, 45 days before or after). When there were several sampling events in a given year, we selected the one that took place at the closest date from the most frequently sampled date of the site. Finally, we excluded 1,340 sites that had been limed as part of the long-term Swedish liming programme (https://kalkdatabasen.lansstyrelsen.se/)[57] to avoid including sites whose environmental conditions had been experimentally manipulated. The data selection resulted in 4,476 fish community time series, totalling 46,932 sampling events, 326,717 species abundance records and 806 freshwater fish species. The median time span of the time series was of 17 (13, 23; 25th quantile, 75th quantile) years, the median completeness (that is, number of annual samplings/time span) was of 55% (38%, 78%) (Extended Data Fig. 1) and the median first study year was 1997 (1992, 2003). The sites were mostly located in the Palaearctic (75%), Nearctic (20%) and Australasia (5%), and distributed across 307 hydrographic basins. Four countries gathered 85% of the sites, namely Great Britain (29%), France (21%), Sweden (18%) and the United States (18%; Supplementary Table 2).

### Community metrics

We assessed community changes in riverine fish communities using several biodiversity facets related to total abundance, species richness, the share of non-native species and community composition (Supplementary Table 2). Total abundance was reported in number of individuals (47.00% of the sampling events), density of individuals per 100 $m^2$ (51.81%), catch per unit effort (CPUE; 1.05%) and Leslie index (0.14%; Supplementary Table 2). Although we selected for strict protocol consistency, 70% or more of the sampling events by unit of abundance did not report sampling effort, preventing us from harmonizing count, abundance density and CPUE[29].

As sampled species richness is a negatively biased estimator of the 'true' species richness, we corrected sampled species richness with the coverage-based rarefaction and extrapolation methodology[58]. The estimated coverage of a sample is positively related to the number of individuals and negatively related to the number of singletons. We fixed the coverage of all samples at 98.5% via rarefaction and extrapolation using the R package iNEXT[59] to make species richness comparable across samples. We did not always have a direct estimate of the number of individuals and number of singletons to compute the sampling coverage, as 51.81% of the abundances were measured as density values and 1.05% as CPUE. In this case, we first divided each species density ($x_i$) by the minimum value in the community and rounded each value to the nearest integer (that is, $x_i' = \mathrm{round}(1/\min(x_i))$, where $x_i'$ is the estimated abundance of the $i$th species) to obtain at least one singleton species, that is, a species with one individual. However, we note that both covered-based and raw species richness estimates were highly

correlated (Spearman's $\rho = 0.97$ for both raw variables and log-transformed ones), and the choice of the metric did not influence our interpretations regarding the patterns and drivers of species richness changes (Extended Data Fig. 7).

The biogeographic origin of the fish species describing whether species were native or introduced to a given drainage basin was retrieved using a global database documenting species status across drainage basins of the world[60] (94.3% of the species occurrences; Supplementary Table 2). For the sites falling outside the river basins provided in the global database, such as for the sites located close to the shore, we used the closest basin within the same country. For species not included in a given drainage basin, we determined the origin of the species at the country scale using FishBase[61] (5.5% of species occurrences). Given the spatial extent of the United States, we completed the global database with the Nonindigenous Aquatic Species database developed by the US Geological Survey (https://nas.er.usgs.gov/) using US states as spatial grain (0.05% of the species occurrences). We completed the remaining species origins at the country scale, using national atlases and FishBase data in neighbouring countries, such as for *Piaractus brachypomus* and *Rutilus rutilus* in the United States (0.1% of the species occurrences; Supplementary Table 2). We then estimated the percentage of non-native species with respect to both abundance and species richness for each sampling event (Supplementary Table 3).

## Dissimilarity metrics

We used the complement of the Jaccard similarity index (that is, Jaccard dissimilarity, which we denote as $J$) to characterize temporal dissimilarity in community composition at each site, taking the first year of sampling of a community as the reference community. This index is based on presence/absence and is simply the sum of species gains and losses over the total number of species across two samples (equation (1)). It thus measures the proportion of species not shared between two samples.

$$J = \frac{S_{\text{gain}} + S_{\text{loss}}}{S_{\text{tot}}} \tag{1}$$

with $S_{\text{gain}}$, $S_{\text{loss}}$ and $S_{\text{tot}}$ being the numbers of immigrant, extirpated and total species, respectively.

We further partitioned the Jaccard dissimilarity index into two complementary indices, turnover ($J_t$) and nestedness ($J_n$), respectively $J_t = (2 \times \min(S_{\text{loss}}, S_{\text{gain}}))/(S_{\text{common}} + (2 \times \min(S_{\text{loss}}, S_{\text{gain}})))$ and $J_n = 1 - J_t$, $S_{\text{common}}$ being the number of species present in both communities[62]. High turnover values indicate that the changes in community composition result from species replacement, whereas high values of nestedness indicate species gains or losses from a nested community, that is, that a community is a subset of the other[63].

We further characterized temporal dissimilarity with the Simpson-based dissimilarity index[24] ($H_d$, equation (2)). This index is based on species relative abundances and their variation across two samples. Simpson-based dissimilarity index is based on the Simpson diversity index and thus gives higher weight to changes in the abundant species, whereas Jaccard dissimilarity index gives equal weight to each species. Simpson-based dissimilarity index thus quantifies the extent of changes in the identity of the dominant species[24]. Both high Jaccard and Simpson dissimilarity values indicate changes in composition resulting from changes in the abundant species, whereas conjointly high Jaccard and low Simpson dissimilarity values indicate composition changes resulting from changes in the species with low relative abundances.

$$H_d = 1 - H$$
$$H_d = 1 - \frac{\sum_i (p_i - p_i')^2}{\sum_i p_i^2 + \sum_i p_i'^2 - \sum_i p_i p_i'} \tag{2}$$

where $i$ is species $i$, $p$ is relative abundance and $'$ is the focal community.

## Environmental drivers

We quantified the degree of anthropogenic pressures using the human footprint index[26,30], extracted from the RiverATLAS database at the reach scale[52,64] (stream segment length average of 450 m). We did so by snapping the sites to the closest stream segment using a 1 km buffer (99% of the sites). The human footprint index aggregates an array of human pressures, including population density and the extent of urban, forested, cropland and pastureland areas, but also transportation hubs such as roads, railways and navigable pathways. It does so by combining remote sensing data, systematic surveys and modelling from ground data, making it less prone to errors[30]. The human footprint index ranges from 0 to 50, with values superior to 4 being considered in a degraded state[41]. This index has been previously related to species extinction and risk of biological invasion[32,41]. To capture both the effects of the legacy of past anthropogenic pressures and of its recent changes, we considered the human footprint index computed in 1993 and 2009 (that is, a 16-year span). Specifically, the human footprint index of 1993 was used as a measure of past anthropogenic pressures at the beginning of the study period, and the ratio between the human footprint of 2009 and 1993 as a measure of the recent changes in anthropogenic pressures (only 7% of the samplings took place before 1993, while 58% took place between 1993 and 2009, and 34% after 2009; Extended Data Fig. 1). We chose the human footprint index developed in ref. 30 because it goes farther back in time than other related human footprint indices (for example, 1993 versus 2000 in ref. 51). Nevertheless, both indices are highly correlated (Spearman's $\rho = 0.81$ for 2009) in overlapping years, suggesting that our results are robust to the choice of human footprint index. In order to obtain interpretable coefficients of recent changes in human footprint, we log-transformed the ratio of human footprint with a base 2, such as values of −1 and 1 represented a division by 2 and a multiplication by 2 of the human footprint between 1993 and 2009, respectively. In river networks, the environmental heterogeneity and connectivity along the longitudinal stream position (upstream–downstream) strongly shape species occurrences, immigration rates and community composition[28]. To capture this longitudinal gradient, we first described stream characteristics at each site by the altitude (m), slope (°), average annual discharge (m³ s⁻¹), distance from source (km) and Strahler order (that is, downstream position based on stream/tributary hierarchy) that we extracted from the RiverATLAS database at the stream segment scale[52,64]. We next performed a PCA over the site stream characteristics on the log-transformed (after adding absolute minimum values plus 1 to avoid few negative values in altitude) and standardized variables (that is, centred and scaled). We orthogonally rotated the two first principal components using the varimax criterion[65], to increase the quality of the variable representation (that is, their loadings) on the two first principal components. The first rotated component was positively related to average annual discharge, distance from source and Strahler order, and captured 56% of the variance (Extended Data Fig. 8), and was subsequently used as a composite variable describing the longitudinal stream position from upstream to downstream (from negative to positive values, respectively). The correlation coefficients indicated little covariation among the predictors (maximum Spearman's correlation of 0.09 found between past and recent pressures; Supplementary Fig. 1).

## Statistical analysis

To estimate community temporal trends, we first modelled the different community metrics ($Y$) as dependent of time ($\beta_0\text{Time}_t$; equation (3)), measured as the number of years since the beginning of the sampling at each site. The statistical model (equation (3)) was adapted according to the nature of the response variable. For total abundance, we added the measurement unit of abundance as a categorical variable both as a main effect and in interaction with time[33]. We set the factor level 'raw count' as the reference level such that the temporal trends in total abundance in the main text and Supplementary Information

are expressed in raw count (that is, number of individuals). For dissimilarity metrics, we set the intercept fixed at 0 as dissimilarity metrics at each site was 0 at the beginning of the time series. We accounted for the hierarchical spatial structure in the data by assigning random effects on the intercept ($\alpha$) and on the slope of the temporal trends ($\beta_0$) conditional on basin identity ($n$) and on site identity ($i$) nested within basin ($i|n$). The random effects and the error terms were modelled as a normal distribution of mean 0 and variance ($\sigma^2$).

$$Y_{i|n,t} = \alpha + \beta_0 \text{Time}_t + \epsilon_{i|n,t} \tag{3}$$

where $\alpha = \alpha_0 + a_n + a_{i|n}, \beta_0 = \mu + b_n + b_{i|n}, \alpha_0$ and $\mu$ being the fixed intercept and slope, $a$ and $b$ being the random intercept and slope, $\epsilon_{i|n,t}$ the residual error, and $a_n, a_{i|n}, b_n, b_{i|n}, \epsilon_{i|n,t} \sim \mathcal{N}(0, \sigma^2)$.

To assess the drivers of community change, we then built a second model incorporating additional covariates ($X_k$, $k$ being the index of the covariate, equation (4)): the degree of past anthropogenic pressures measured by the human footprint index of 1993, the recent changes in anthropogenic pressures measured by the ratio between the human footprint index of 2009 and 1993, and the longitudinal stream position estimated by the rotated PCA axis. We included two-way interactions between time and the ecological drivers ($\sum_{k=1} \beta_{0k} \text{Time}_t X_k$) to test how longitudinal stream position and anthropogenic pressures affect the temporal trends in community metrics. For instance, a positive interaction between time and past anthropogenic pressures would indicate that faster changes were happening in historically degraded areas. We further included the three-way interactions between time and the pairs of other ecological drivers ($\sum_{k=1,k\neq l} \beta_{0kl} \text{Time}_t X_k X_l$, $k$ and $l$ being the indexes of the covariates) to examine the presence of synergistic or antagonistic effects between anthropogenic pressures and longitudinal stream position on the temporal trends in community metrics (that is, the results in Fig. 3a). For instance, a positive interaction between time, recent changes in anthropogenic pressures and longitudinal stream position would indicate that the effects of recent changes in anthropogenic pressures on temporal trends were more important in downstream areas. We included all the predictors as main effects ($\sum_{k=1} \beta_k X_k$) to capture the effects of the ecological drivers on the spatial variation in average community composition metrics through time, except when modelling the dissimilarity metrics (that is, we only included the effects of ecological drivers on the temporal trends, the results in Fig. 3b). We did so because dissimilarity metrics quantify community changes at a given time point ($t = 0...N$) at a given site from the first year ($t = 0$) and therefore bounded by 0 and 1 at each site; we thus did not expect average differences in dissimilarity related to any other factors than time. We did not include interactions among ecological drivers on the spatial part of the model as we restricted the core of the analysis to the drivers of temporal rather than spatial variation in freshwater fish community composition and tried to keep the model as parsimonious as possible. Finally, we derived the comparison in community metrics across space according to the ecological drivers by using the predictions of the model at the baseline ($t = 0$). The predictions controlled for the values of other predictors (such as longitudinal stream position and past anthropogenic pressures) by setting them at their median values.

$$Y_{i|n,t} = \alpha + \beta_0 \text{Time}_t + \sum_{k=1} \beta_k X_k + \sum_{k=1} \beta_{0k} \text{Time}_t X_k$$
$$+ \sum_{k=1,k\neq l} \beta_{0kl} \text{Time}_t X_k X_l + \epsilon_{i|n,t} \tag{4}$$

where $k$, $l$ and $\in$ [1, 2, 3] are ecological drivers including stream gradient, legacy of past and recent changes in anthropogenic pressures.

All the response variables were modelled with a Gaussian distribution following previous studies modelling temporal trends in total abundance, species richness and community composition at the global scale[2,3,33]. Other error structures might be more appropriate to model

response variables bounded between 0 and 1 or ratio of discrete numbers such as the dissimilarity metrics and the proportion of non-native species. However, doing so allowed us to obtain easily interpretable coefficients across all community metrics (for example, temporal trends cannot be interpreted as rates of change when modelled using a logit scale such as when using a beta distribution). In addition, a previous study using similar models[3] found that slope coefficients estimated with a Gaussian error and a beta error had a Spearman correlation superior to 0.90 and gave qualitatively similar results. We therefore believe that this choice is not likely to alter our conclusions.

We log-transformed the number of years as log(year + 1) as it improved the quality of the model fitting to the data, decreasing the Watanabe–Akaike information criterion[66,67] by −733 on average (−11%) across community metrics facets (Supplementary Table 4). It suggested the presence of nonlinearity in the temporal trends, which is particularly expected in the case of bounded variables such as the dissimilarity metrics. We log-transformed total abundance and species richness, so that their temporal trends are multiplicative and can be expressed in percentage change by unit of time. We then derived the percentage of change by decade in species richness and total abundance by back transforming $\beta_0$ using a time value of log(10 + 1) as follows: $(e^{\beta_0 \times \log(10+1)} - 1) \times 100$.

In order to compare the strength of the effects among predictors across community metrics, we scaled both community metrics and the predictors by their standard deviation prior to model fitting. As our models contain interactions, the individual slope coefficients could be difficult to interpret without centring the predictors around ecological relevant values[68]. As an example, the average temporal trends estimated by $\beta_0$ in equation (4) can only be interpreted when all the $X_k = 0$. Hence, we centred past anthropogenic pressures and longitudinal stream position around their average values. The variables quantifying time and recent changes in anthropogenic pressures were not centred, because then the main effects of the ecological predictors ($\sum_{k=1} \beta_k$) would indicate their baseline effects (that is, when time is equal to 0 and without recent changes in anthropogenic pressures).

The models were evaluated in a Bayesian framework using integrated nested Laplacian approximation[69,70] (INLA), which approximates the posterior distribution of the parameters and does not rely on Markov chains and Monte Carlo simulations, and thus is a computationally efficient method to evaluate Bayesian models. We computed the CI at 80%, 90% and 95% using the highest posterior density method[71], which can respectively be interpreted as weak, moderate and strong evidence of an effect when the interval does not include 0 (refs. 33,34). The statistical models were implemented using the INLA R package[69], with defaults uninformative priors. The prior distribution of fixed coefficients followed a flat zero centred normal distribution ($\mathcal{N}(\mu, \sigma^2) = \mathcal{N}(0, 1000)$). The prior distribution of the random effects and the gaussian error ($\epsilon_{it}$, eq. (3)) followed a log gamma distribution with shape ($s$) and inverse scale ($\tau$) parameters ($\mathcal{G}(s, \tau) = \mathcal{G}(1, 5.10^{-5})$). We then back-transformed the estimated coefficients to the standard deviations attributed to the random effects and the gaussian error ($\sigma = 1/\sqrt{\tau}$). We checked that the slope coefficients, random effects and the temporal trends by basin and site were similar to those obtained with an implementation in frequentist. Then, we concluded that the quality of parameter inference did not suffer from the uninformative priors.

We checked the overall quality of the model fit to the data by plotting the fitted versus observed values (Supplementary Fig. 2). We visually inspected the posterior integral transform and conditional predictive ordinate distribution to assess both the quality of data representation and the frequency of outliers. There was very little multicollinearity in the model, as all variance inflation factors were around 1 (Supplementary Table 5).

We computed marginal ($R_m^2$) and conditional ($R_c^2$) $R^2$ (equation (5)) to assess the quality of the fit of the Bayesian models, respectively

associated with the variance explained by the fixed effects and the variance explained by both the fixed and random effects[72]. We only included the random effects on the intercept in the $R^2$ computation, that is, the basin effect ($a_n$) and the site effect ($a_{i|n}$), as the inclusion of the variance attributed to random slopes is much more complex and was shown to not change the results[72,73]. We computed the variance associated with each predicted value (Var$_{fit}$) from their posterior distribution, following recommendations to take into account the variability associated to the priors[74]. We then reported the mean marginal and conditional $R^2$ associated their 95% CI computed using the highest posterior density method.

$$R_m^2 = \frac{\text{Var}_{fit}}{\text{Var}_{fit}+\text{Var}_{res}} = \frac{\sigma^2(\hat{y}_i)}{\sigma^2(\hat{y}_i)+\sigma^2(y_i-\hat{y}_i)}$$
$$R_c^2 = \frac{\text{Var}_{fit}+(a_n)^2+(a_{i|n})^2}{\text{Var}_{fit}+(a_n)^2+(a_{i|n})^2+\text{Var}_{res}}$$
$$(5)$$

with $y_i$ and $\hat{y}_i$ being respectively the observations and the predicted values, and Var$_{fit}$ and Var$_{res}$ being respectively the variance of the predictive means and the variance of the residuals[74]. $a_n$ and $a_{i|n}$ are respectively the standard deviation on the random intercept associated with the hydrographic basin and the site within the basin.

## Typology of temporal trends

To estimate the covariations among multiple dimensions of community change, we performed a PCA on the temporal trends in the community metrics estimated at each site with the models (equation (3)) having time as a sole fixed predictor (that is, using the predictions of the models in percentage per decade using the best linear unbiased prediction method). We used four variables in this analysis: temporal trends in total species richness, community abundance, temporal dissimilarity and turnover. We excluded the two variables describing the non-native species composition (proportion of non-native abundance and richness) because their predicted temporal trends at the site level displayed a heavy tailed distribution compared with the other variables (kurtosis of the distributions: 10 versus 21 and 8 versus 15 for total and non-native abundance and species richness, respectively; Fig. 1), which in turn exerted a disproportionate constraint on the analysis. In addition, we performed a clustering analysis on the temporal trends in the community metrics at the site level to identify distinct types of community trajectory using the trimmed $k$-means method[75], a robust clustering method because it avoids the identification of spurious clusters. The method consists of trimming the most outlying data in the multidimensional space, the number of dimensions being the number of community metrics. To choose a relevant number of clusters, we plotted the log-likelihood of the trimmed classification as a function of the proportion of the most outlying data trimmed ($\alpha$) and the number of clusters (Supplementary Fig. 3). We thus selected a partition of temporal community changes in six clusters with $\alpha = 5\%$ (see Supplementary Fig. 4 with four clusters). We did not constrain the algorithm for the relative size or shape of the clusters, as we had no a priori expectation about them. The clustering algorithm was run for a minimum of 100 iterations and up to a maximum of 125 iterations. To further control for the quality of the cluster assignment, we discarded any fish community for which the second-best cluster assignment was 50% better than the first one by comparing the degree of affiliation to the clusters[75]. The clustering was performed using the tclust R package[75].

## Reporting summary

Further information on research design is available in the Nature Portfolio Reporting Summary linked to this article.

## Data availability

The data used in the study are open access, although we provide the raw data to facilitate the reproduction of the analysis (https://doi.org/10.5281/zenodo.7817360).

## Code availability

The manuscript and the Supplementary Information are written in R Markdown, that is, combining code and text, and are available on GitHub (https://github.com/alaindanet/RivFishTimeBiodiversity-Facets). We further implemented a code pipeline using the targets R package to ensure that the code, data, figures, manuscript and results are up to date.

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

## Acknowledgements

We thank J. Näslund for his assistance accessing the data from the long-term Swedish liming programme. We also thank all the monitoring programme crews and researchers for their continuous efforts of data collection and for making these data publicly available; this study would not be possible without their efforts and generosity. In particular, we would like to acknowledge the authors that gave us the permission to use the data or who made their data available for this study (Supplementary Table 1). We further thank S. Bonnamour, C. Fontaine, G. Loïs, M. Mouchet and E. Porcher at the Centre d'Écologie et des Sciences de la Conservation, as well as A. P. Beckerman and T. F. Johnson at the University of Sheffield for their very constructive feedbacks on the study. J.D.O. was supported by the Richard C. and Lois M. Worthington Endowed Professor in Fisheries Management from the School of Aquatic and Fishery Sciences, University of Washington.

## Author contributions

All authors carried out conceptualization and methodology. A.D. carried out software, validation, formal analysis, visualization and wrote the original draft. X.G. and J.D.O. carried out writing—review and editing. L.C. carried out data curation, supervision, project management, funding acquisition and writing—review and editing.

## Competing interests

The authors declare no competing interests.

## Additional information

**Extended data** is available for this paper at https://doi.org/10.1038/s41559-023-02271-x.

**Correspondence and requests for materials** should be addressed to Alain Danet.

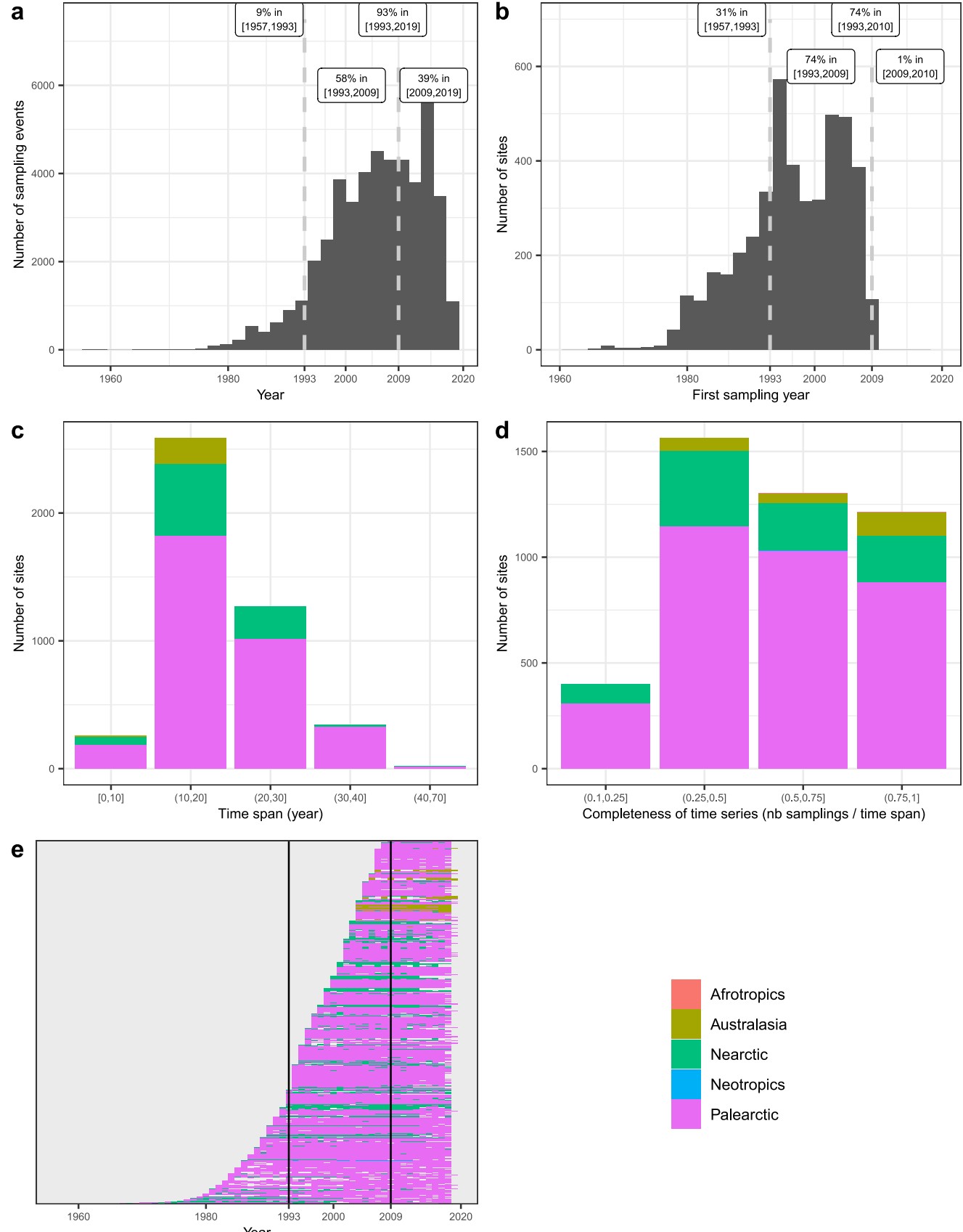

**Extended Data Fig. 1 | Characteristics of the fish community time series.**
**a**, Distribution of the year of all sampling events. **b**–**d**, Distribution of the (**b**) first year of sampling, (**c**) time span, and (**d**) completeness of the time series across sites. e, time series of yearly sampling events. Colours correspond to the biogeographic realms. In (**a**), (**b**), and (**e**), the lines display the year 1993 and 2009. The years 1993 and 2009 correspond to the years of the human footprint index assessments, which were used to quantify past and recent changes in anthropogenic pressures (See Methods).

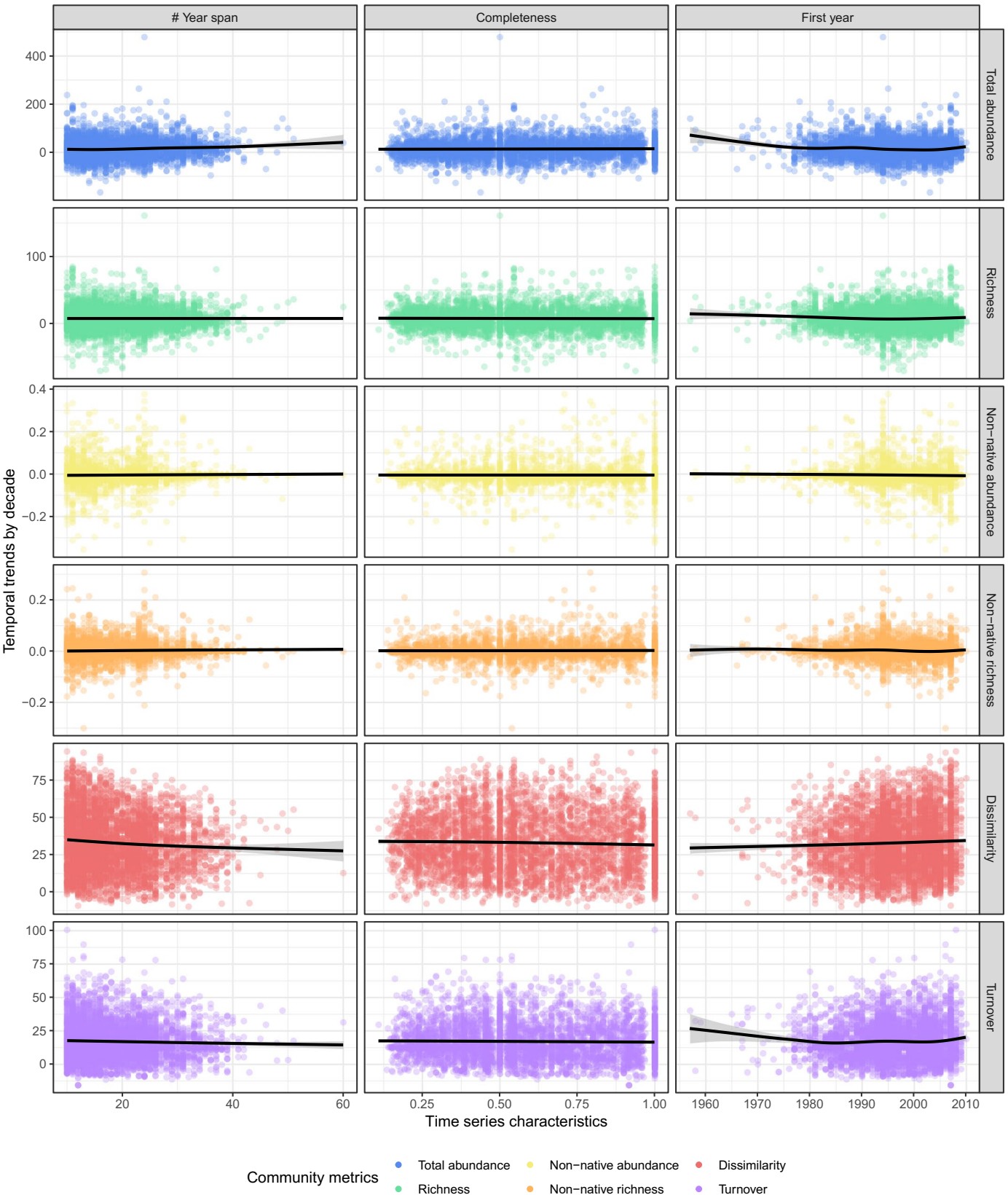

**Extended Data Fig. 2 | Covariations between temporal trends estimated at the site level and the characteristics of the time series.** The year span is expressed in number of years, completeness in proportion of years over the entire study period and first year of sampling in year. The temporal trends were estimated with time as sole fixed predictor (See Methods, eq. (3)). The black lines display the regression lines fitted with a Generalized Additive Model (GAM) and the gray area the confidence interval.

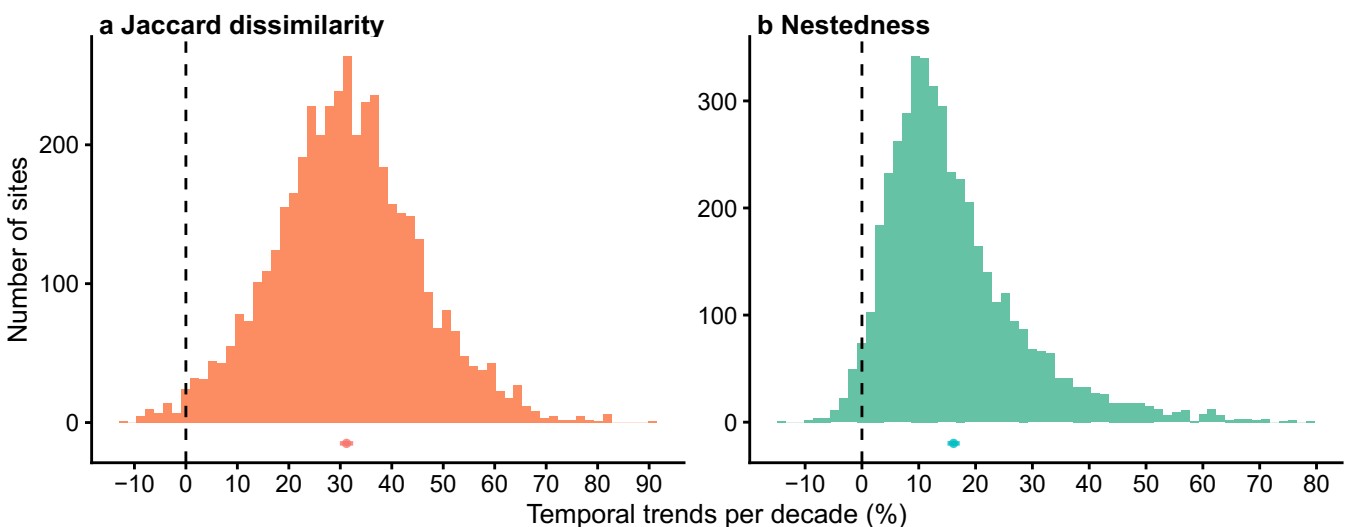

**Extended Data Fig. 3 | Distribution of community temporal trends across sites.** Temporal trends per decade in Jaccard dissimilarity (**a**) and nestedness (**b**). The temporal trends were estimated from a hierarchical Bayesian model including time as sole predictor (See Methods). The histograms show the Best Linear Unbiased Predictor (BLUP) estimated at each site. The points depict the average posterior distribution with bars depicting the Bayesian credible intervals at 95%. The dashed lines refer to no temporal trend. N = 46,932 sampling events across 4,476 sites.

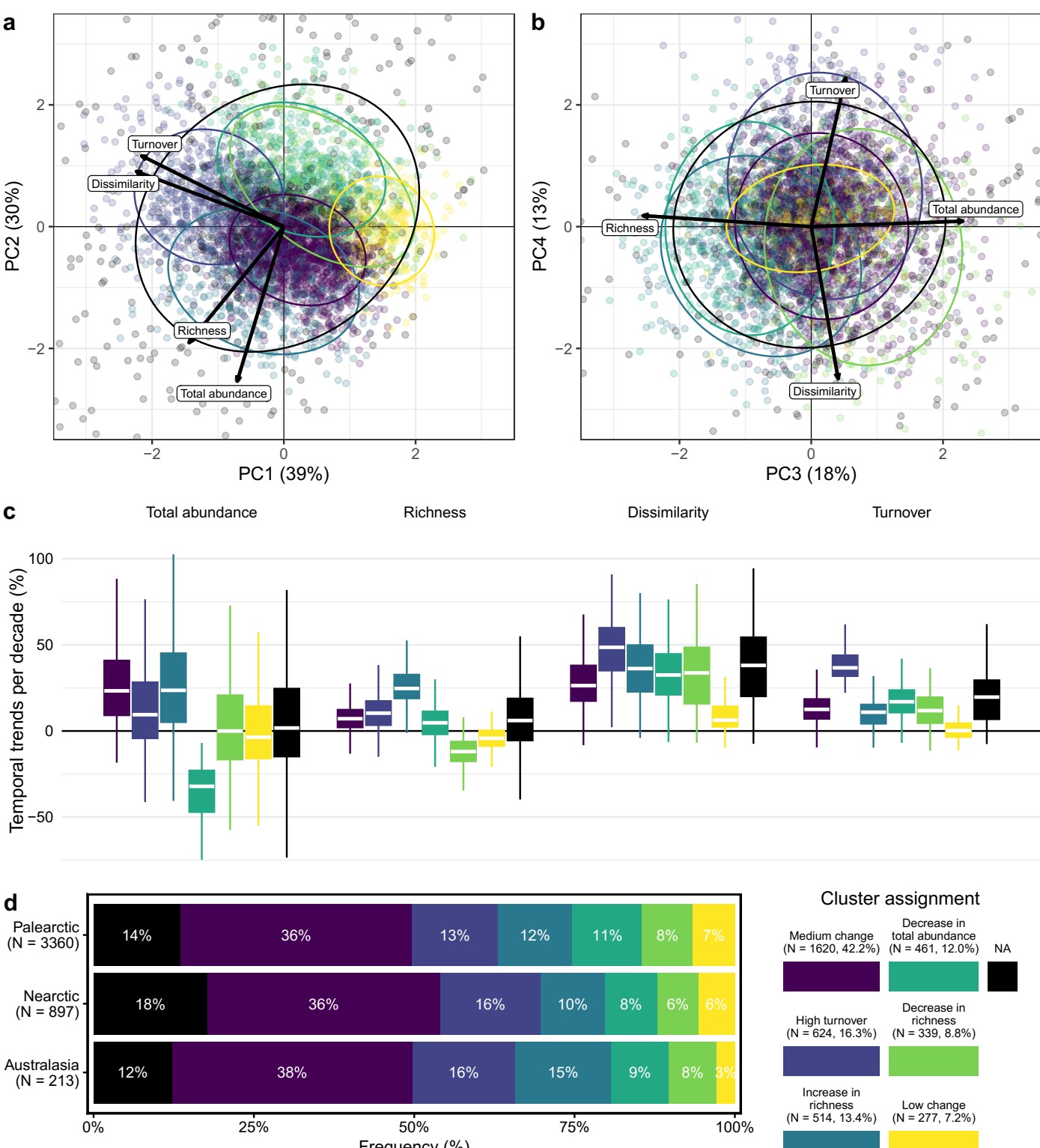

**Extended Data Fig. 4 | Covariation among the community temporal trends and characterization of community trajectories.** (a, b) PCA biplot of the community temporal trends and their cluster assignment where the sites are colored according to their cluster assignment. (c) Boxplots displaying the distribution of the temporal trends by cluster. The center of the box depicts the median while the bounds depicts the 25% and 75% percentile. The whiskers depict the extreme values within 1.5 interquartile range beyond the bounds of the box. (d) Cluster frequencies across the three main biogeographic realms.

The ellipses in (a-b) display the 95% intervals around the clusters assuming a Student's t distribution. The clusters were named according t o their main characteristics. Contrary to Fig. 2 (Main text), we included the sites whose cluster affiliation was uncertain ('NA', black color, (N = 641, 14%). NA clusters are well distributed over the PCA (a-b), which is confirmed by the distribution of community metric temporal trends (c). We observe a few more NAs in the Nearctic realm (+5%, that is around 40 sites over 897).

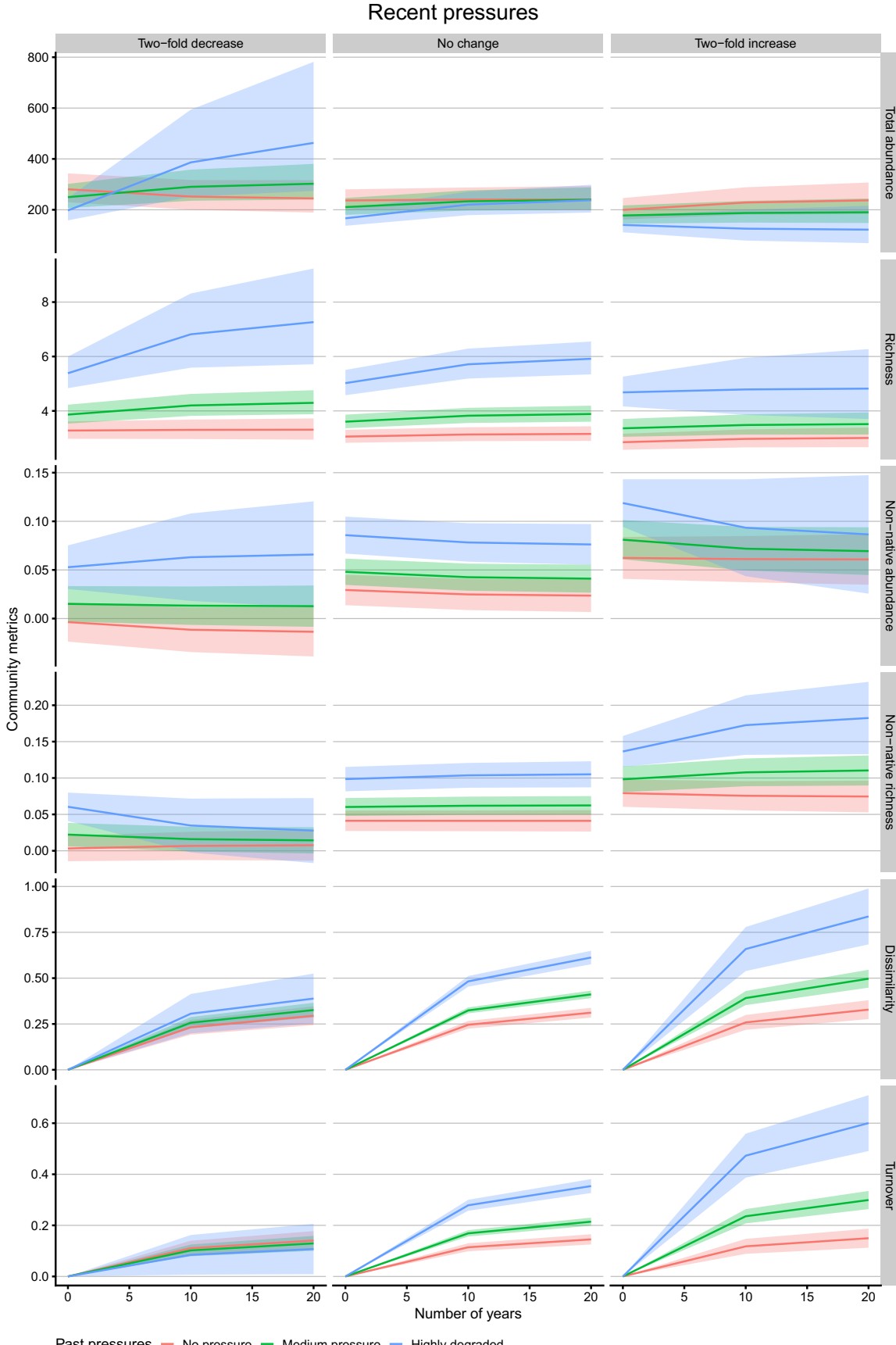

**Extended Data Fig. 5 | See next page for caption.**

**Extended Data Fig. 5 | Predictions of community metric changes over time according to past human pressures and recent changes in human pressures using a hierarchical Bayesian model.** Total abundance is expressed in count, species richness in number of species (raw), while non-native abundance, non-native richness, dissimilarity and turnover are expressed in proportion. No pressure, medium pressure, and highly degraded past pressure levels correspond respectively to values of human footprint index for the year 1993 of 2.5 (intact ecosystem), 16.8 (median value in our dataset), and 45.6 (maximum value in the dataset). Recent pressure levels represent change in human footprint index between the years 1993 and 2009 (see Methods for details). The central lines display the mean, while the lower and upper lines display the credible intervals at 95% of the average posterior distribution.

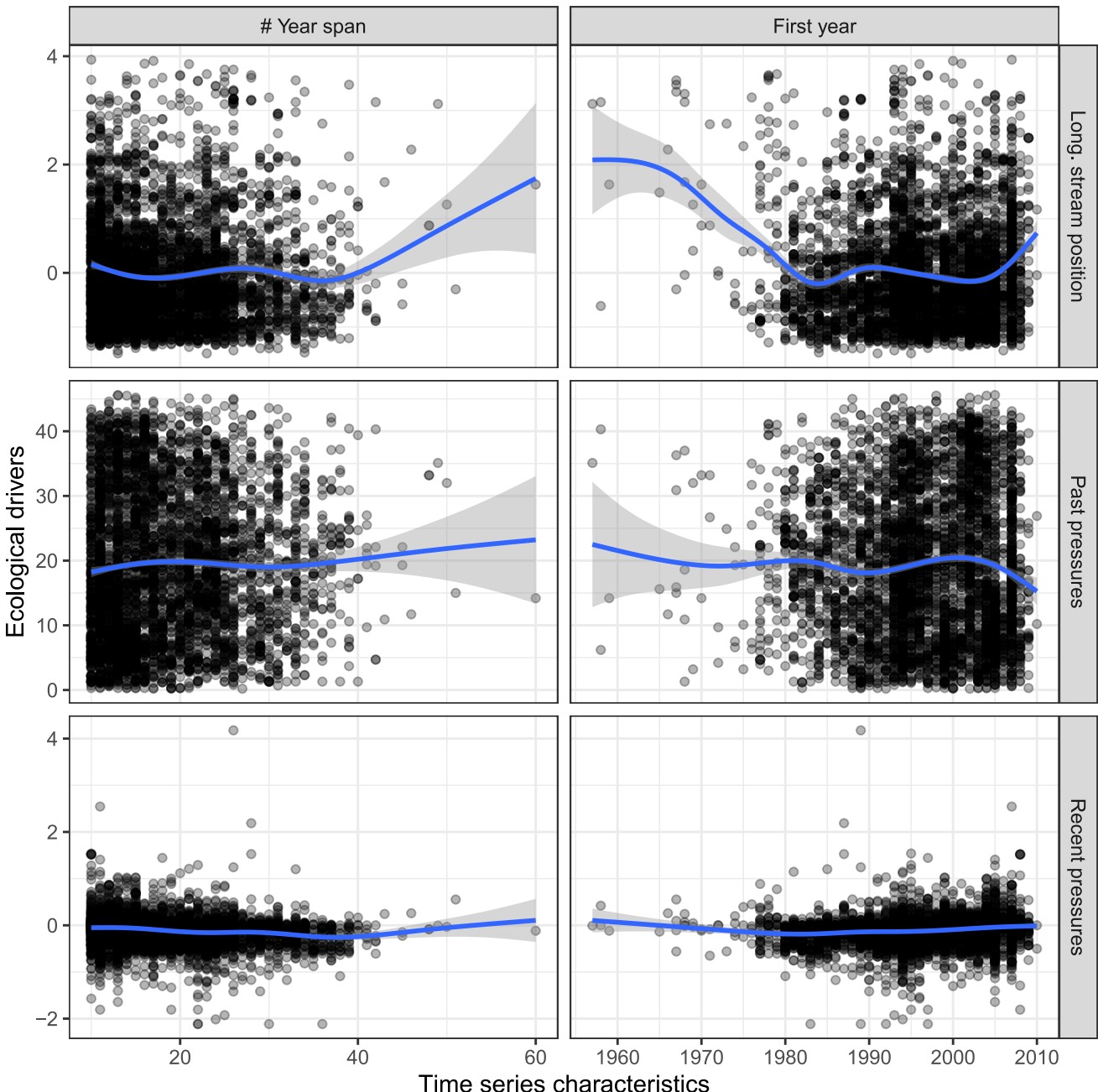

**Extended Data Fig. 6 | Covariations between ecological drivers used in the hierarchical Bayesian model (longitudinal stream position, past and recent changes in anthropogenic pressures) and time series characteristics (time span in number of years and beginning of the time series in year).** The blue lines display the regression lines fitted with a Generalized Additive Model (GAM) and the gray area the confidence interval. The ecological drivers were largely unrelated to the characteristics of the time series (N = 4,476 sites).

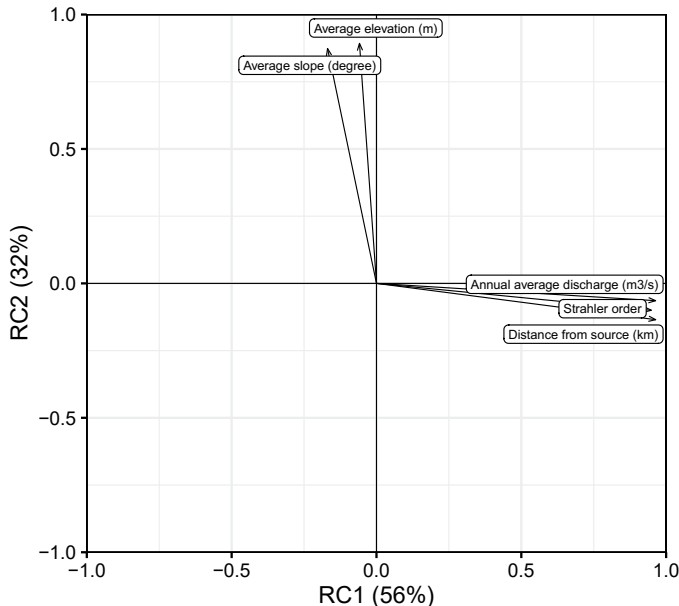

**Extended Data Fig. 7 | Rotated PCA over the physical and hydrological characteristics of the sites.** PCA axes (Rotated Component, RC) were rotated using the varimax algorithm (See Methods). The first axis is related to the average discharge, Strahler order and the distance from source, and was subsequently used in the statistical modelling as a composite variable intended to summarise the longitudinal stream position from upstream to downstream.

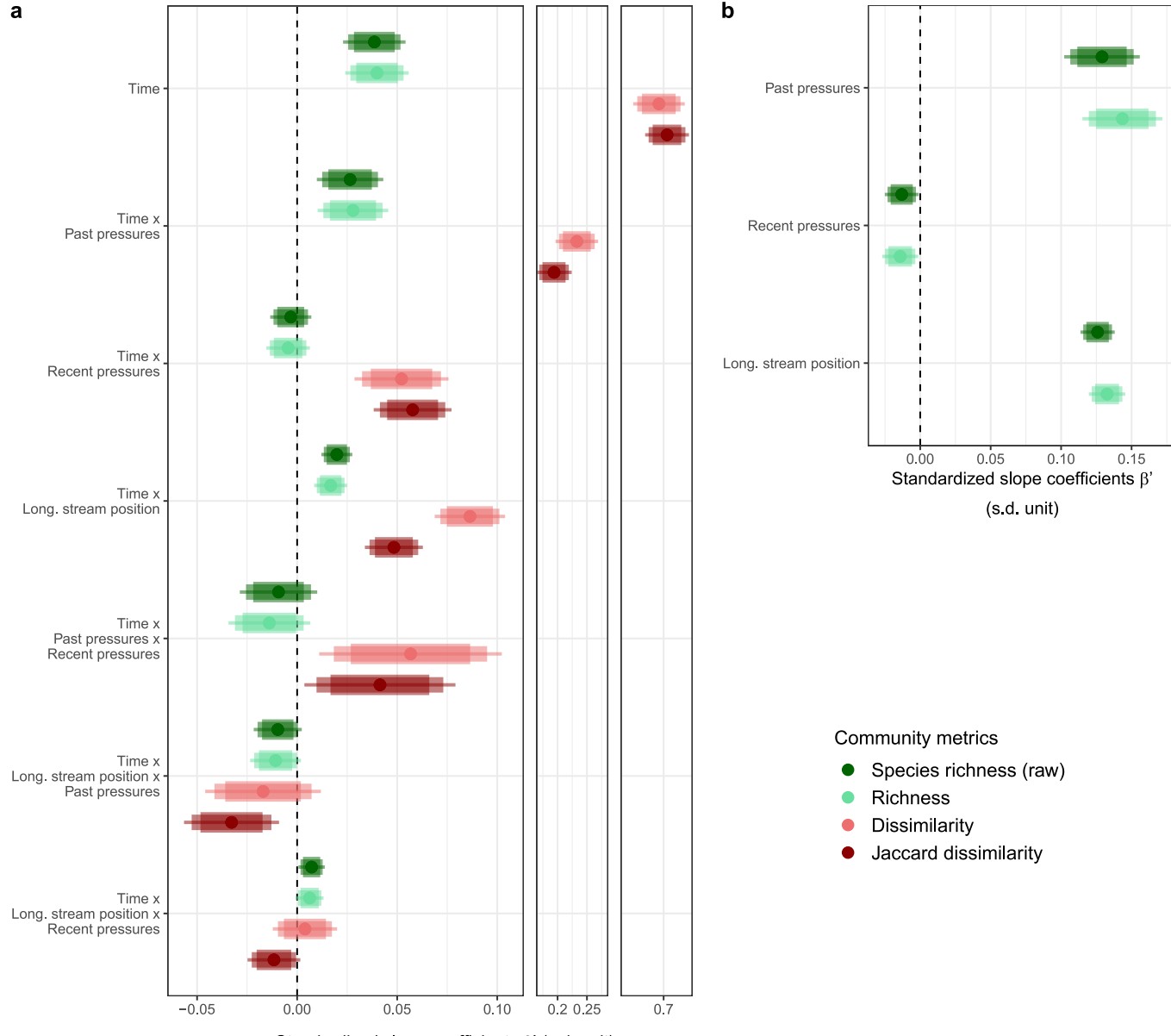

**Extended Data Fig. 8 | Drivers of temporal change and spatial variation in fish community metrics. a,b,** Effects of anthropogenic pressures and longitudinal (long.) stream position on temporal changes (**a**) and spatial variation (**b**) in fish community metrics. See Fig. 3 in the main text for details. We compare coverage corrected species richness (Richness) with raw species richness (Richness (raw)) and Simpson dissimilarity (Dissimilarity) with Jaccard dissimilarity (Jaccard dissimilarity). We observe that using coverage based species richness or raw species richness does not affect the results. Similarly, we observe consistent effect sizes for Simpson and Jaccard dissimilarity indices, meaning that observed changes in community composition concerned abundant species and thus changes in the identity of dominant species (see Methods). The points depict the average posterior distribution. Large, medium and thin bars depict the Bayesian credible intervals at 80, 90 and 95%, respectively. Please note the broken abscissa scale in (a). N = 46,932 sampling events across 4,476 sites.

**Extended Data Table 1 | Random effects associated with the estimation of the temporal trends computed from the model containing only time as fixed predictor**

| Response variable | Temporal trends s.d. | |
| --- | --- | --- |
| | Time (basin) | Time (site nested in basin) |
| Abundance (total) | 0.167 [0.149,0.195] | 0.225 [0.211,0.239] |
| Richness | 0.082 [0.074,0.091] | 0.103 [0.098,0.11] |
| Non-native abundance | 0.009 [0.006,0.013] | 0.027 [0.026,0.028] |
| Non-native richness | 0.008 [0.007,0.009] | 0.021 [0.02,0.022] |
| Dissimilarity | 0.038 [0.033,0.047] | 0.091 [0.088,0.094] |
| Turnover | 0.031 [0.025,0.038] | 0.069 [0.065,0.074] |

[95% CI]: Credible Interval computed using the Highest Posterior Density method.

# Reporting Summary

## Statistics

For all statistical analyses, confirm that the following items are present in the figure legend, table legend, main text, or Methods section.

| n/a | Confirmed | |
|---|---|---|
| ☐ | ☒ | The exact sample size (*n*) for each experimental group/condition, given as a discrete number and unit of measurement |
| ☐ | ☒ | A statement on whether measurements were taken from distinct samples or whether the same sample was measured repeatedly |
| ☐ | ☒ | The statistical test(s) used AND whether they are one- or two-sided *Only common tests should be described solely by name; describe more complex techniques in the Methods section.* |
| ☐ | ☒ | A description of all covariates tested |
| ☐ | ☒ | A description of any assumptions or corrections, such as tests of normality and adjustment for multiple comparisons |
| ☐ | ☒ | A full description of the statistical parameters including central tendency (e.g. means) or other basic estimates (e.g. regression coefficient) AND variation (e.g. standard deviation) or associated estimates of uncertainty (e.g. confidence intervals) |
| ☒ | ☐ | For null hypothesis testing, the test statistic (e.g. *F*, *t*, *r*) with confidence intervals, effect sizes, degrees of freedom and *P* value noted *Give P values as exact values whenever suitable.* |
| ☐ | ☒ | For Bayesian analysis, information on the choice of priors and Markov chain Monte Carlo settings |
| ☐ | ☒ | For hierarchical and complex designs, identification of the appropriate level for tests and full reporting of outcomes |
| ☒ | ☐ | Estimates of effect sizes (e.g. Cohen's *d*, Pearson's *r*), indicating how they were calculated |

*Our web collection on statistics for biologists contains articles on many of the points above.*

## Software and code

Policy information about availability of computer code

| Data collection | The study did not use data collection |
|---|---|
| Data analysis | The code pipeline to perform the data analysis is available on github: https://github.com/alaindanet/RivFishTimeBiodiversityFacets |

For manuscripts utilizing custom algorithms or software that are central to the research but not yet described in published literature, software must be made available to editors and reviewers. We strongly encourage code deposition in a community repository (e.g. GitHub). See the Nature Portfolio guidelines for submitting code & software for further information.

## Data

Policy information about availability of data

All manuscripts must include a data availability statement. This statement should provide the following information, where applicable:

- Accession codes, unique identifiers, or web links for publicly available datasets
- A description of any restrictions on data availability
- For clinical datasets or third party data, please ensure that the statement adheres to our policy

The origin of the dataset,and the links to retrieve them are available in Table S2. All the raw files used to conduct the analysis are available on Zenodo: https://doi.org/10.5281/zenodo.7817360

## Research involving human participants, their data, or biological material

Policy information about studies with human participants or human data. See also policy information about sex, gender (identity/presentation), and sexual orientation and race, ethnicity and racism.

| | |
|---|---|
| Reporting on sex and gender | *Use the terms sex (biological attribute) and gender (shaped by social and cultural circumstances) carefully in order to avoid confusing both terms. Indicate if findings apply to only one sex or gender; describe whether sex and gender were considered in study design; whether sex and/or gender was determined based on self-reporting or assigned and methods used. Provide in the source data disaggregated sex and gender data, where this information has been collected, and if consent has been obtained for sharing of individual-level data; provide overall numbers in this Reporting Summary. Please state if this information has not been collected. Report sex- and gender-based analyses where performed, justify reasons for lack of sex- and gender-based analysis.* |
| Reporting on race, ethnicity, or other socially relevant groupings | *Please specify the socially constructed or socially relevant categorization variable(s) used in your manuscript and explain why they were used. Please note that such variables should not be used as proxies for other socially constructed/relevant variables (for example, race or ethnicity should not be used as a proxy for socioeconomic status). Provide clear definitions of the relevant terms used, how they were provided (by the participants/respondents, the researchers, or third parties), and the method(s) used to classify people into the different categories (e.g. self-report, census or administrative data, social media data, etc.) Please provide details about how you controlled for confounding variables in your analyses.* |
| Population characteristics | *Describe the covariate-relevant population characteristics of the human research participants (e.g. age, genotypic information, past and current diagnosis and treatment categories). If you filled out the behavioural & social sciences study design questions and have nothing to add here, write "See above."* |
| Recruitment | *Describe how participants were recruited. Outline any potential self-selection bias or other biases that may be present and how these are likely to impact results.* |
| Ethics oversight | *Identify the organization(s) that approved the study protocol.* |

Note that full information on the approval of the study protocol must also be provided in the manuscript.

# Field-specific reporting

Please select the one below that is the best fit for your research. If you are not sure, read the appropriate sections before making your selection.

☐ Life sciences  ☐ Behavioural & social sciences  ☒ Ecological, evolutionary & environmental sciences

For a reference copy of the document with all sections, see nature.com/documents/nr-reporting-summary-flat.pdf

# Life sciences study design

All studies must disclose on these points even when the disclosure is negative.

| | |
|---|---|
| Sample size | *Describe how sample size was determined, detailing any statistical methods used to predetermine sample size OR if no sample-size calculation was performed, describe how sample sizes were chosen and provide a rationale for why these sample sizes are sufficient.* |
| Data exclusions | *Describe any data exclusions. If no data were excluded from the analyses, state so OR if data were excluded, describe the exclusions and the rationale behind them, indicating whether exclusion criteria were pre-established.* |
| Replication | *Describe the measures taken to verify the reproducibility of the experimental findings. If all attempts at replication were successful, confirm this OR if there are any findings that were not replicated or cannot be reproduced, note this and describe why.* |
| Randomization | *Describe how samples/organisms/participants were allocated into experimental groups. If allocation was not random, describe how covariates were controlled OR if this is not relevant to your study, explain why.* |
| Blinding | *Describe whether the investigators were blinded to group allocation during data collection and/or analysis. If blinding was not possible, describe why OR explain why blinding was not relevant to your study.* |

# Behavioural & social sciences study design

All studies must disclose on these points even when the disclosure is negative.

| | |
|---|---|
| Study description | *Briefly describe the study type including whether data are quantitative, qualitative, or mixed-methods (e.g. qualitative cross-sectional, quantitative experimental, mixed-methods case study).* |
| Research sample | *State the research sample (e.g. Harvard university undergraduates, villagers in rural India) and provide relevant demographic information (e.g. age, sex) and indicate whether the sample is representative. Provide a rationale for the study sample chosen. For studies involving existing datasets, please describe the dataset and source.* |

| Sampling strategy | *Describe the sampling procedure (e.g. random, snowball, stratified, convenience). Describe the statistical methods that were used to predetermine sample size OR if no sample-size calculation was performed, describe how sample sizes were chosen and provide a rationale for why these sample sizes are sufficient. For qualitative data, please indicate whether data saturation was considered, and what criteria were used to decide that no further sampling was needed.* |
| --- | --- |
| Data collection | *Provide details about the data collection procedure, including the instruments or devices used to record the data (e.g. pen and paper, computer, eye tracker, video or audio equipment) whether anyone was present besides the participant(s) and the researcher, and whether the researcher was blind to experimental condition and/or the study hypothesis during data collection.* |
| Timing | *Indicate the start and stop dates of data collection. If there is a gap between collection periods, state the dates for each sample cohort.* |
| Data exclusions | *If no data were excluded from the analyses, state so OR if data were excluded, provide the exact number of exclusions and the rationale behind them, indicating whether exclusion criteria were pre-established.* |
| Non-participation | *State how many participants dropped out/declined participation and the reason(s) given OR provide response rate OR state that no participants dropped out/declined participation.* |
| Randomization | *If participants were not allocated into experimental groups, state so OR describe how participants were allocated to groups, and if allocation was not random, describe how covariates were controlled.* |

# Ecological, evolutionary & environmental sciences study design

All studies must disclose on these points even when the disclosure is negative.

| Study description | We investigated how past and recent anthropogenic pressures shape spatial variation and temporal trends of riverine fish communities in the last decades. |
| --- | --- |
| Research sample | We used RIvFishTime (Comte et al. 2021), a dataset compiling more than 12,000 time series containing species abundances of riverine fish communities. |
| Sampling strategy | N/A |
| Data collection | N/A |
| Timing and spatial scale | The median first sampling year of the timeseries was 1997 (1992-2003, resp. 25% and 75% quartile (Fig. S1). The median time span was 17 years (12-23) and the median completness of the timeseries was 58% (38% - 78%). The sites were mostly located in the Palearctic (75%), Nearctic (20%) and Australasia (5%). Four countries gathered 85% of the sites, namely Great Britain (29%), France (21%), Sweden (18%), and the USA (18%, Table S3). |
| Data exclusions | We selected timeseries that had at least 5 datapoints and a minimum of 10 year time span (pre-established criteria). We also selected sampling events that had a consistent sampling protocol. we excluded 1,340 sites that had been limed as part of the long-term Swedish liming program (available at https://kalkdatabasen.lansstyrelsen.se/)49 to avoid including sites whose environmental conditions had been experimentally manipulated. The data selection resulted in 4,476 fish community time series, totalling 46,932 sampling events, 326,717 species abundance records. |
| Reproducibility | We used the "targets" R package ro ensure that the analysis pipeline was up to date. The code is available at https://github.com/alaindanet/RivFishTimeBiodiversityFacets. |
| Randomization | N/A |
| Blinding | N/A |

Did the study involve field work?   ☐ Yes   ☒ No

# Field work, collection and transport

| Field conditions | *Describe the study conditions for field work, providing relevant parameters (e.g. temperature, rainfall).* |
| --- | --- |
| Location | *State the location of the sampling or experiment, providing relevant parameters (e.g. latitude and longitude, elevation, water depth).* |
| Access & import/export | *Describe the efforts you have made to access habitats and to collect and import/export your samples in a responsible manner and in compliance with local, national and international laws, noting any permits that were obtained (give the name of the issuing authority, the date of issue, and any identifying information).* |
| Disturbance | *Describe any disturbance caused by the study and how it was minimized.* |

# Reporting for specific materials, systems and methods

We require information from authors about some types of materials, experimental systems and methods used in many studies. Here, indicate whether each material, system or method listed is relevant to your study. If you are not sure if a list item applies to your research, read the appropriate section before selecting a response.

## Materials & experimental systems

| n/a | Involved in the study |
|---|---|
| ☐ ☐ | Antibodies |
| ☐ ☐ | Eukaryotic cell lines |
| ☐ ☐ | Palaeontology and archaeology |
| ☐ ☐ | Animals and other organisms |
| ☐ ☐ | Clinical data |
| ☐ ☐ | Dual use research of concern |
| ☐ ☐ | Plants |

## Methods

| n/a | Involved in the study |
|---|---|
| ☐ ☐ | ChIP-seq |
| ☐ ☐ | Flow cytometry |
| ☐ ☐ | MRI-based neuroimaging |

## Antibodies

| Antibodies used | *Describe all antibodies used in the study; as applicable, provide supplier name, catalog number, clone name, and lot number.* |
|---|---|
| Validation | *Describe the validation of each primary antibody for the species and application, noting any validation statements on the manufacturer's website, relevant citations, antibody profiles in online databases, or data provided in the manuscript.* |

## Eukaryotic cell lines

Policy information about cell lines and Sex and Gender in Research

| Cell line source(s) | *State the source of each cell line used and the sex of all primary cell lines and cells derived from human participants or vertebrate models.* |
|---|---|
| Authentication | *Describe the authentication procedures for each cell line used OR declare that none of the cell lines used were authenticated.* |
| Mycoplasma contamination | *Confirm that all cell lines tested negative for mycoplasma contamination OR describe the results of the testing for mycoplasma contamination OR declare that the cell lines were not tested for mycoplasma contamination.* |
| Commonly misidentified lines (See ICLAC register) | *Name any commonly misidentified cell lines used in the study and provide a rationale for their use.* |

## Palaeontology and Archaeology

| Specimen provenance | *Provide provenance information for specimens and describe permits that were obtained for the work (including the name of the issuing authority, the date of issue, and any identifying information). Permits should encompass collection and, where applicable, export.* |
|---|---|
| Specimen deposition | *Indicate where the specimens have been deposited to permit free access by other researchers.* |
| Dating methods | *If new dates are provided, describe how they were obtained (e.g. collection, storage, sample pretreatment and measurement), where they were obtained (i.e. lab name), the calibration program and the protocol for quality assurance OR state that no new dates are provided.* |

☐ Tick this box to confirm that the raw and calibrated dates are available in the paper or in Supplementary Information.

| Ethics oversight | *Identify the organization(s) that approved or provided guidance on the study protocol, OR state that no ethical approval or guidance was required and explain why not.* |
|---|---|

Note that full information on the approval of the study protocol must also be provided in the manuscript.

## Animals and other research organisms

Policy information about studies involving animals; ARRIVE guidelines recommended for reporting animal research, and Sex and Gender in Research

| Laboratory animals | *For laboratory animals, report species, strain and age OR state that the study did not involve laboratory animals.* |
|---|---|

| Wild animals | *Provide details on animals observed in or captured in the field; report species and age where possible. Describe how animals were caught and transported and what happened to captive animals after the study (if killed, explain why and describe method; if released, say where and when) OR state that the study did not involve wild animals.* |
| Reporting on sex | *Indicate if findings apply to only one sex; describe whether sex was considered in study design, methods used for assigning sex. Provide data disaggregated for sex where this information has been collected in the source data as appropriate; provide overall numbers in this Reporting Summary. Please state if this information has not been collected. Report sex-based analyses where performed, justify reasons for lack of sex-based analysis.* |
| Field-collected samples | *For laboratory work with field-collected samples, describe all relevant parameters such as housing, maintenance, temperature, photoperiod and end-of-experiment protocol OR state that the study did not involve samples collected from the field.* |
| Ethics oversight | *Identify the organization(s) that approved or provided guidance on the study protocol, OR state that no ethical approval or guidance was required and explain why not.* |

Note that full information on the approval of the study protocol must also be provided in the manuscript.

# Clinical data

Policy information about clinical studies

All manuscripts should comply with the ICMJE guidelines for publication of clinical research and a completed CONSORT checklist must be included with all submissions.

| Clinical trial registration | *Provide the trial registration number from ClinicalTrials.gov or an equivalent agency.* |
| Study protocol | *Note where the full trial protocol can be accessed OR if not available, explain why.* |
| Data collection | *Describe the settings and locales of data collection, noting the time periods of recruitment and data collection.* |
| Outcomes | *Describe how you pre-defined primary and secondary outcome measures and how you assessed these measures.* |

# Dual use research of concern

Policy information about dual use research of concern

## Hazards

Could the accidental, deliberate or reckless misuse of agents or technologies generated in the work, or the application of information presented in the manuscript, pose a threat to:

No | Yes
- ☐ | ☐ Public health
- ☐ | ☐ National security
- ☐ | ☐ Crops and/or livestock
- ☐ | ☐ Ecosystems
- ☐ | ☐ Any other significant area

## Experiments of concern

Does the work involve any of these experiments of concern:

No | Yes
- ☐ | ☐ Demonstrate how to render a vaccine ineffective
- ☐ | ☐ Confer resistance to therapeutically useful antibiotics or antiviral agents
- ☐ | ☐ Enhance the virulence of a pathogen or render a nonpathogen virulent
- ☐ | ☐ Increase transmissibility of a pathogen
- ☐ | ☐ Alter the host range of a pathogen
- ☐ | ☐ Enable evasion of diagnostic/detection modalities
- ☐ | ☐ Enable the weaponization of a biological agent or toxin
- ☐ | ☐ Any other potentially harmful combination of experiments and agents

# Plants

| | |
|---|---|
| Seed stocks | *Report on the source of all seed stocks or other plant material used. If applicable, state the seed stock centre and catalogue number. If plant specimens were collected from the field, describe the collection location, date and sampling procedures.* |
| Novel plant genotypes | *Describe the methods by which all novel plant genotypes were produced. This includes those generated by transgenic approaches, gene editing, chemical/radiation-based mutagenesis and hybridization. For transgenic lines, describe the transformation method, the number of independent lines analyzed and the generation upon which experiments were performed. For gene-edited lines, describe the editor used, the endogenous sequence targeted for editing, the targeting guide RNA sequence (if applicable) and how the editor was applied.* |
| Authentication | *Describe any authentication procedures for each seed stock used or novel genotype generated. Describe any experiments used to assess the effect of a mutation and, where applicable, how potential secondary effects (e.g. second site T-DNA insertions, mosiacism, off-target gene editing) were examined.* |

# ChIP-seq

## Data deposition

☐ Confirm that both raw and final processed data have been deposited in a public database such as GEO.

☐ Confirm that you have deposited or provided access to graph files (e.g. BED files) for the called peaks.

| | |
|---|---|
| Data access links<br>*May remain private before publication.* | *For "Initial submission" or "Revised version" documents, provide reviewer access links.  For your "Final submission" document, provide a link to the deposited data.* |
| Files in database submission | *Provide a list of all files available in the database submission.* |
| Genome browser session<br>(e.g. UCSC) | *Provide a link to an anonymized genome browser session for "Initial submission" and "Revised version" documents only, to enable peer review.  Write "no longer applicable" for "Final submission" documents.* |

## Methodology

| | |
|---|---|
| Replicates | *Describe the experimental replicates, specifying number, type and replicate agreement.* |
| Sequencing depth | *Describe the sequencing depth for each experiment, providing the total number of reads, uniquely mapped reads, length of reads and whether they were paired- or single-end.* |
| Antibodies | *Describe the antibodies used for the ChIP-seq experiments; as applicable, provide supplier name, catalog number, clone name, and lot number.* |
| Peak calling parameters | *Specify the command line program and parameters used for read mapping and peak calling, including the ChIP, control and index files used.* |
| Data quality | *Describe the methods used to ensure data quality in full detail, including how many peaks are at FDR 5% and above 5-fold enrichment.* |
| Software | *Describe the software used to collect and analyze the ChIP-seq data. For custom code that has been deposited into a community repository, provide accession details.* |

# Flow Cytometry

## Plots

Confirm that:

☐ The axis labels state the marker and fluorochrome used (e.g. CD4-FITC).

☐ The axis scales are clearly visible. Include numbers along axes only for bottom left plot of group (a 'group' is an analysis of identical markers).

☐ All plots are contour plots with outliers or pseudocolor plots.

☐ A numerical value for number of cells or percentage (with statistics) is provided.

## Methodology

| | |
|---|---|
| Sample preparation | *Describe the sample preparation, detailing the biological source of the cells and any tissue processing steps used.* |
| Instrument | *Identify the instrument used for data collection, specifying make and model number.* |
| Software | *Describe the software used to collect and analyze the flow cytometry data. For custom code that has been deposited into a community repository, provide accession details.* |

| Cell population abundance | *Describe the abundance of the relevant cell populations within post-sort fractions, providing details on the purity of the samples and how it was determined.* |
| --- | --- |
| Gating strategy | *Describe the gating strategy used for all relevant experiments, specifying the preliminary FSC/SSC gates of the starting cell population, indicating where boundaries between "positive" and "negative" staining cell populations are defined.* |

☐ Tick this box to confirm that a figure exemplifying the gating strategy is provided in the Supplementary Information.

# Magnetic resonance imaging

## Experimental design

| Design type | *Indicate task or resting state; event-related or block design.* |
| --- | --- |
| Design specifications | *Specify the number of blocks, trials or experimental units per session and/or subject, and specify the length of each trial or block (if trials are blocked) and interval between trials.* |
| Behavioral performance measures | *State number and/or type of variables recorded (e.g. correct button press, response time) and what statistics were used to establish that the subjects were performing the task as expected (e.g. mean, range, and/or standard deviation across subjects).* |

## Acquisition

| Imaging type(s) | *Specify: functional, structural, diffusion, perfusion.* |
| --- | --- |
| Field strength | *Specify in Tesla* |
| Sequence & imaging parameters | *Specify the pulse sequence type (gradient echo, spin echo, etc.), imaging type (EPI, spiral, etc.), field of view, matrix size, slice thickness, orientation and TE/TR/flip angle.* |
| Area of acquisition | *State whether a whole brain scan was used OR define the area of acquisition, describing how the region was determined.* |

Diffusion MRI     ☐ Used     ☐ Not used

## Preprocessing

| Preprocessing software | *Provide detail on software version and revision number and on specific parameters (model/functions, brain extraction, segmentation, smoothing kernel size, etc.).* |
| --- | --- |
| Normalization | *If data were normalized/standardized, describe the approach(es): specify linear or non-linear and define image types used for transformation OR indicate that data were not normalized and explain rationale for lack of normalization.* |
| Normalization template | *Describe the template used for normalization/transformation, specifying subject space or group standardized space (e.g. original Talairach, MNI305, ICBM152) OR indicate that the data were not normalized.* |
| Noise and artifact removal | *Describe your procedure(s) for artifact and structured noise removal, specifying motion parameters, tissue signals and physiological signals (heart rate, respiration).* |
| Volume censoring | *Define your software and/or method and criteria for volume censoring, and state the extent of such censoring.* |

## Statistical modeling & inference

| Model type and settings | *Specify type (mass univariate, multivariate, RSA, predictive, etc.) and describe essential details of the model at the first and second levels (e.g. fixed, random or mixed effects; drift or auto-correlation).* |
| --- | --- |
| Effect(s) tested | *Define precise effect in terms of the task or stimulus conditions instead of psychological concepts and indicate whether ANOVA or factorial designs were used.* |

Specify type of analysis:     ☐ Whole brain     ☐ ROI-based     ☐ Both

| Statistic type for inference<br><br>(See Eklund et al. 2016) | *Specify voxel-wise or cluster-wise and report all relevant parameters for cluster-wise methods.* |
| --- | --- |
| Correction | *Describe the type of correction and how it is obtained for multiple comparisons (e.g. FWE, FDR, permutation or Monte Carlo).* |

## Models & analysis

| n/a | Involved in the study |
|---|---|
| ☐ | ☐ Functional and/or effective connectivity |
| ☐ | ☐ Graph analysis |
| ☐ | ☐ Multivariate modeling or predictive analysis |

**Functional and/or effective connectivity**
*Report the measures of dependence used and the model details (e.g. Pearson correlation, partial correlation, mutual information).*

**Graph analysis**
*Report the dependent variable and connectivity measure, specifying weighted graph or binarized graph, subject- or group-level, and the global and/or node summaries used (e.g. clustering coefficient, efficiency, etc.).*

**Multivariate modeling and predictive analysis**
*Specify independent variables, features extraction and dimension reduction, model, training and evaluation metrics.*

