## [Peer Review File · Nature Ecology & Evolution]

Peer Review Information

Journal: Nature Ecology & Evolution

Manuscript Title: Past and recent anthropogenic pressures drive rapid changes in riverine fish communities

Corresponding author name(s): Alain Danet

Editorial Notes:

Reviewer Comments & Decisions:

Decision Letter, initial version:

7th July 2023

Dear Dr Danet,

Your manuscript entitled "Past and recent anthropogenic pressures drive rapid changes in riverine fish communities" has now been seen by 3 reviewers, whose comments are attached. The reviewers have raised a number of concerns which will need to be addressed before we can offer publication in Nature Ecology & Evolution. We will therefore need to see your responses to the criticisms raised and to some editorial concerns, along with a revised manuscript, before we can reach a final decision regarding publication.

We therefore invite you to revise your manuscript taking into account all reviewer and editor comments. Please highlight all changes in the manuscript text file.

* If you have not done so already please begin to revise your manuscript so that it conforms to our Article format instructions at <http://www.nature.com/natecolevol/info/final-submission>. Refer also to any guidelines provided in this letter.

2[REDACTED]

Nature Ecology & Evolution is committed to improving transparency in authorship. As part of our efforts in this direction, we are now requesting that all authors identified as 'corresponding author' on published papers create and link their Open Researcher and Contributor Identifier (ORCID) with their account on the Manuscript Tracking System (MTS), prior to acceptance. ORCID helps the scientific community achieve unambiguous attribution of all scholarly contributions. You can create and link your ORCID from the home page of the MTS by clicking on 'Modify my Springer Nature account'. For more information please visit www.springernature.com/orcid.

[REDACTED]

Reviewer expertise:

Reviewer #1: global freshwater ecology and biodiversity, HFI, community ecology

Reviewer #2: human land use/human footprint index, freshwater ecology

Reviewer #3: biodiversity trends, human impact

Reviewers' comments:

Reviewer #1 (Remarks to the Author):

My assessment is included in the document attached.

Reviewer #2 (Remarks to the Author):

2This study presented extremely interesting research on how past and recent human pressure has shaped fish biodiversity (richness, abundance, and composition) over time. A massive database is employed across a large spatial scale, making the results remarkably consistent. I find the results of this study of broad interest, especially to aquatic ecologists and fisheries managers. I found no major flaws in the structure and analysis of the manuscript. Although I do not consider the conclusion to be extremely new, they add very important information for understanding fish biodiversity changes and how human pressures contribute to such changes. I have some major doubts and suggestions that I believe will improve the clarity of the study.

While I understand that the journal has word limitations, I think the authors could, in a few sentences in the last paragraph of the introduction, explain what was considered human pressure (in this case, human footprint). Particularly, what is the Human Footprint? and which metrics of human pressure layers this index considers? Still, would it be possible to draw the human footprint map: past (1997) and recent (2009), and make these two maps available, including the surveyed sites in the supplemental material? The HFP map is easily drawn with the coordinates of the sampling sites. I believe that a map of the HFP in 1993 and 2009 could improve the reader's visualization of the change in HFP over time.

I have a concern/doubt about the structure of Bayesian models, especially the interaction between human footprint and time. Overall, human footprint and time are not independent; that is, the human footprint scale with time <https://doi.org/10.1038/s41597-022-01284-8>. It is also confusing how the interaction between human footprint vs time was considered if there are only two temporal measures of human footprint (1993 and 2009). For example, how was Time interacted with Past pressures if there was only Human footprint measurement in 1993? Time seems to be a category with only 1 level for the human footprint, how can we interpret such an interaction? How could the HFP interact with Time if there is no temporal replicability of the HFP? If the authors really want to test the temporal effect of HFP, it might be necessary to use annual HFP values. There are annual HFP measurements from 2000 to 2018 (see: <https://doi.org/10.1038/s41597-022-01284-8>). If the intention is just to compare the effect of past and present HFP on biodiversity, perhaps the HFP vs. Time interaction is unnecessary since it is just comparing the slope of past and present HFP. I really don't know if the interaction between time and human footprint is necessary, since this was not discussed in the paper. In any case, a little more explanation of the interaction between HFP vs. Time should be provided. It appears that the authors have switched the colors of the proportion of non-native richness and abundance in Fig. 3 and in the text. For example, in the text, the authors state that the effect of past anthropogenic pressures on the proportion of non-native richness is displayed in yellow, and on the proportion of non-native abundance is displayed in orange (page 10: second paragraph). However, in Fig. 3 proportion of non-native richness is displayed in orange while the proportion of non-native abundance is displayed in yellow. Please, fix this.

The slope of the interaction between Time \times Stream long. position \times past pressures on the proportion of non-native abundance do not overlap with zero (Fig. 3a). Doesn't this indicate a strong effect of the past anthropogenic pressure on the non-native abundance in most downstream areas (following the interpretation of 80% = weak, 90% = moderate, and 95% = strong)? If yes, consider changing the "moderate evidence" to "strong evidence" in the text.

Although remarkably consistent, this study is correlational. I missed some consideration of this in the discussion or elsewhere in the manuscript.

3Page 9: What do you mean by "higher degree of past anthropogenic pressures"? Perhaps it is better: "increase in past anthropogenic pressures".

Page 15 (second paragraph). It is stated that: "The increase in local species richness over time could thus result from the introduction of non-native species". Doesn't this statement contradict the results of Fig. 1, in which the richness of non-native species does not change and the abundance of invasive species decreases over time? Fig. 1 (c and d) seems to show no increase in the richness and abundance of non-native fish species over time. Perhaps you mean: "The increase in local species richness in highly degraded rivers over time could thus result from the introduction of non-native species".

Reviewer #3 (Remarks to the Author):

This manuscript uses time series of river fish assemblages to assess how various metrics of biodiversity differ in space and over time and in relation to past and recent human impact. The paper finds change in community composition and increases in both abundance and richness over the time series assessed. They also assess how the proportion of non-native species richness and abundance has changed over time, which could explain some of the positive trends in abundance and richness. This was a valuable addition as positive trends in freshwater biodiversity are often contested, going beyond the usual metrics to assess change in more detail.

I found the content of the manuscript interesting, I thought the interactive map was a nice touch, and it was very impressive that the document was written in R Markdown.

There are however a number of things that I think need to be addressed before the work can be published within this journal.

1. Limited description of the data and methods in the main text.

I found it very challenging to fully understand what the data included and how the analyses were carried out from reading the main text of this manuscript. In those journals where the methods are after the rest of the main text, I think it is really important to be able to get a good understanding of what the data are, and briefly, how they were assessed.

It is really unclear, for example, what the time series data represent, where they are from, how they were collected and the general coverage from the information given in the introduction and results. The final paragraph of the introduction could include much more information on the data, data coverage, and analysis as well as linking to the specific analyses of interest.

Things that could be made more clear in particular – the last 3 decades is used consistently to describe the coverage, but the data used when you get to the method extends further back than this is some cases; across three biogeographic realms – descriptions of the data present three main regions, but then the map presents data from elsewhere.

42. Sensitivity tests and other checks

The authors describe their methods used in relation to other studies that have followed a similar analytical framework, often stating tests that these other papers cover, however there very few explorations or tests carried out on the data used here to provide reassurance for the presented results. Some questions that came to my mind when reading included: Does the length of the time series impact the results (i.e. are longer time series more likely to be increasing? Since they have had longer to recover from past impact. Do time series that were collected in more recent years or that start further back have an influence on the overall results? Is there any bias in terms of geographic region? Does each region cover similar time frames, lengths of time series, range of predictors? Is there an influence of start year on clustering? E.g. Are earlier/later time series associated with certain types of change? It would be reassuring to see the authors address some points to explore the potential influence of certain subsets of the data, particularly give the criticisms that many of the previous studies that this one cites within the methods have faced.

Minor comments:

There were no line numbers so apologies if it is not clear where these comments relate to.

Abstract:

The description of the data is very vague. "across three biogeographic realms", it would be good to list them here. Ideally, this should also give some idea of the timescale over which trends are being assessed.

Introduction:

Rather than "the last 3 decades", can you be specific about data coverage.

Repeatedly sampled over the last 3 decades – sounds like all time series have the same start/end and duration. Alternatively; "repeatedly sampled for variable durations and frequencies between 19XX and 20XX."

Expect to see something about how human pressures have changed over the time period assessed, perhaps specifically in relation to where the data are from. Since most data are from developed northern hemisphere, are most regions likely to be improving in terms of water quality at least?

Not clear to me what is meant by longitudinal stream position. There are also a number of terms that seem to be specialist for the freshwater realm throughout the that would benefit from explanation (reach, Stahler order)

No information on how the data were analysed. Suggest something starting like " We used Bayesian hierarchical models to assess change in fish abundance, richness... as a function of time and of ...".

Results:

Community temporal trends

5In the first paragraph, the first set of results are repeated, the first instance can be removed.

“strong evidence” and switching between 80/90/95% CIs. Since this is not general practise and not often seen in papers, I think it might be best either to explain this approach in the main text, not just in the methods, or to stick to use of one set of CI ranges. In the second paragraph of the results, no values are given but just “(strong evidence)” supplied in brackets. If the methods have not been read first, I think this could be potentially confusing to a reader. Suggest removing the (strong evidence) either way.

Paragraph 3 – time as a random slope, but Fig 1 legend says time as the only predictor. It is not clear from the current explanation what the models actually represent without having to go to the methods. Description so far in the main text indicates models include time with random effects of basin and site only.

Typology of community temporal trends

Starting description could be altered to briefly describe the method and how/why it was used before going into the results. At the moment, this section doesn't flow well in terms of what was done and why.

Figure 2 legend, description of cluster names might be helpful here, especially for medium change which is not very clear without reading the methods.

Drivers of community temporal trends

What is Strahler order, needs explanation for wide audience journal.

Non-native species abundance referred to as orange in the figure but it is actually yellow.

Figure S4 – repeat of non-native abundance. Think one should be richness. Also line and ribbon plot might be easier to visualise than 3 lines here.

Drivers of community variation across space

It is not clear what the model is that determines these results. Please make it clear what variables are being assessed in each instance.

Do these models account for year of sampling at all?

Most upstream rather than upstream-most.

Discussion

Why do you think the results here differ from the global trends?

What about the time point chosen for the past? What impact could this have on results? Especially considering most of the data come from regions where pressures have been reducing in freshwater systems compared to further back in the past. There are certainly some points around baseline and reference points that could be discussed and potentially explored with some additional tests.

Methods

The description of the data suggest data from certain regions only, then the interactive map shows other regions are included. Please be clear about where the data are from. I think it would be useful to have a map of the sites in the main text.

What were the data sampling, were they sampling everything, certain species? Very little information to help understand the data and the potential biases.

The quantiles are presented for the data, I think ranges would be more informative.

The description of the hierarchical nature of the data, sites within basins (?) could be improved.

Dissimilarity – first year as reference, but first year varies between time series. Could this influence the results?

The methods section (and linked to the final paragraph of the introduction actually) would benefit from clearer relation of questions/areas being explored to the models being run.

*****END*****

Author Rebuttal to Initial comments

Response to reviewers

Legend:

- Response
- *Previous text*
- New text

The response letter is accompanied by a document highlighting the changes introduced in the manuscript, the lines and page numbers are referring to this document.

Reviewer 1

A long-standing question in the Anthropocene is how ecological communities will respond to new environmental conditions influenced by human activities. The current manuscript bridges the knowledge gap by (i) examining the temporal dynamics of community composition, (ii) characterizing these temporal dynamics and (iii) identifying the main drivers of changes including geospatial and human footprint indices. The authors address these issues by utilizing long-term (~30 years) biomonitoring data on stream fish from three distinct biogeographical regions. This data builds upon previously published global-scale fish data. While previous research has explored temporal dynamics in communities, as referenced by the authors in the main text, the novelty of this current work lies in its extensive number of sampling sites, comprehensive spatial and temporal coverage, and investigation of both legacy and current anthropogenic effects on freshwater fish.

Results demonstrated slight increases in abundance and species richness, while indicating significant changes in community composition throughout the study period. Additionally, the temporal dynamics of the community appear to be influenced by past and present anthropogenic pressures, as well as the position of the streams and their interactions at both spatial and temporal scales. The findings suggest a reorganization of fish communities over the last decades with far-reaching consequences for biological conservation and ecosystem health in freshwaters.

I particularly enjoy reading this manuscript and would like to extend my congratulations to the authors for their thorough work in developing a scientifically rigorous and smoothly flowing narrative on the temporal dynamics of ecological communities in global change research. The manuscript is well written, well-structured, the data is well balanced, and the statistical analyses align perfectly with the proposed hypotheses. I don't have any major reservation, but I provide a list of numbered-line comments per page that I would like to see it incorporated in the manuscript or at least discussed here by the authors. I believe that these concerns will not compromise the core message but instead they may help to reinforce the ideas and take-home message on this outstanding work.

We thank the reviewer for their positive feedback on the manuscript and their insightful suggestions that helped to improve the manuscript. We carefully addressed the specific comments below.

1. Page 3 Lines 14–16: I suggest providing a sentence with a specific example of how it is essential to study legacy effects. For instance, Strayer et al. (2006) highlights the need to adopt a long-term perspective on the effects of invasive species to avoid misinterpretation of their impacts.

We agree and thank the reviewer for the reference. It is now cited as follows:

Previous text (P3 - L 48-52): Community composition and trajectories are influenced by past anthropogenic pressures that can generate transient ecological dynamics and long-lasting biotic 'legacies'²⁵. Given the high spatio-temporal heterogeneity of anthropogenic pressures²⁶, ignoring the long-term antecedent effects of historical pressures and their recent changes can greatly impede our understanding of the drivers of community change, such as what has been demonstrated for the effects of invasive species²⁷.

2. Page 9, Line 5: I guess the authors wanted to mention Figure 2a since the two groups turnover-dissimilarity vs richness-abundance appear to be perpendicular (orthogonal) to each other.

We apologize for this confusion. The revised text now reads:

P9 - L 115-118: Temporal trends in community composition (i.e. temporal dissimilarity and turnover) were positively associated with each other as were temporal trends in total abundance and species richness; however these two sets of trajectories appeared largely independent to each other (Fig. 2a).

3. Page 9, last line: Readers from other backgrounds beyond stream ecology may hardly understand this definition. It may be better to state simply that higher values are associated with more downstream parts.

We agree. We have incorporated this suggestion.

P10 - L 139-145: We also uncovered evidence for an interaction with the longitudinal stream position (i.e. represented by a synthetic PCA axis based on several hydromorphological characteristics where high values are associated with more downstream areas, see Methods), such as the legacy effects of past anthropogenic pressures on total abundance (strong evidence) and species richness (weak evidence) were buffered in more downstream areas (resp. CI 95%: $\beta' = -0.021 [-0.031, -0.011]$ and CI 80%: $\beta' = -0.0104 [-0.0188, -0.0021]$, Fig. 3a).

4. Page 10, line 16: Based on Figure S5, it appears that the results are opposed for the Jaccard dissimilarity. There seems to be a significant decrease in Jaccard dissimilarity, which may be more pronounced in upstream parts.

The reviewer is correct. As Simpson-based dissimilarity gives more weight to abundant species than Jaccard dissimilarity, it indicates that a higher degree of anthropogenic pressures in the most downstream areas is associated with greater changes in abundance compared to rare species. We added a sentence to highlight this result.

P10 - L 154-160: We also found that a higher degree of past anthropogenic pressures were associated with faster rates of increases in temporal dissimilarity and turnover (resp. CI 95%: β' = 0.23 [0.20,0.27] in red and 0.25 [0.21,0.29] in purple, Fig. 3a), irrespective of the longitudinal stream position (resp. CI 80%: β' = -0.017 [-0.036, 0.002] and -0.016 [-0.038, 0.005], Fig. 3a). This result was overall consistent between Jaccard and Simpson-based dissimilarity but the effects of past anthropogenic pressures were attenuated in most downstream areas for Jaccard's dissimilarity, indicating that those changes involved more dominant species (Fig. S7).

5. Page 13, line 1: Could you please provide an explanation as to why the authors did not consider the interaction in the spatial models?

Despite initially considering more complex models, after much consideration we opted to not include interactions in the spatial part of the model because it would add 12 additional coefficients to interpret and include in Figure 3. Furthermore, the primary goal of the study was centered on understanding the drivers of temporal rather than spatial variation in freshwater fish community composition. We have now added a sentence to the Methods to explain this decision.

P23 - L 459-465: We did so because dissimilarity metrics quantify the community change at a given time point ($t = 0 \dots N$) at a given site from the first year ($t = 0$) and therefore bounded by 0 and 1 at each site; we thus did not expect average differences in dissimilarity related to any other factors than time. We did not include interactions among ecological drivers on the spatial part of the model as we restricted the core of the analysis to the drivers of temporal rather than spatial variation in freshwater fish community composition and tried to keep the model as parsimonious as possible.

6. Page 13, last lines: Can the authors be more specific about this result? Would it indicate that the fine scale resolutions may report higher predictive power than coarse scales?

The reviewer is correct. We detailed our interpretation further.

P13 - L 211-216: Noteworthy, a larger share of the variance in the community metrics was explained by site and basin identity rather than by the fixed effects alone (R^2 conditional varying from 0.15 for turnover to 0.80 for species richness vs. R^2 marginal varying from 0.02 for species richness to 0.07 for community turnover; Table 1). This indicates that context-dependencies are well captured by our hierarchical models but suggests that integrating fine scale local drivers may further improve our ability to predict local community changes.

7. Page 15, penultimate line: The word "increased" is repeated twice.

We apologize for this typo, we have corrected the revised manuscript.

8. Page 18, line 4: What does it mean “the mode” as being the most frequent protocol?

The reviewer is correct. However, we believe that this methodological jargon was somewhat redundant with the literal explanation, therefore we removed it.

P18 - L 321-326: As several sites had been sampled using different sampling methods (e.g. electrofishing, seining, Table S3), we selected for each site only the sampling events that were performed using the most frequent sampling method. To minimize the influence of seasonal variation in community composition (e.g. due to spawning or migration), we further only selected sampling events within 1.5 months of the most frequently sampled month (i.e. 45 days before or after).

9. Page 18, line 4–5: Is there any reason for selecting a period of 45 days? Perhaps this has been done to avoid the influence of interannual effects, such as spawning or migration, or to maintain similar environmental conditions.

The reviewer is correct. This selection was done to minimize interannual variation in community composition due to seasonal effects that involve the mechanisms listed by the reviewer. We clarified the purpose of this selection in the text (see response to comment #8).

10. Page 18, lines 11–13: How might this affect the overall results? It may be worth considering in the models the number of occurrences, percentage of completeness, and temporal range variation per site to avoid any issues arising from sampling efforts (i.e., communities that have sampled more can have better fittings) and differences in the temporal resolution (i.e., communities with many time gaps can cause unexpected trajectories).

Yes, we agree that in addition to the careful curation of data that we performed, it is important to also check that variation in time series characteristics does not affect the results. To address this question, we checked how temporal trends in community metrics at the site level were affected by the time span (number of years), the completeness and the first year of the time series. None of those descriptors showed a clear association with the temporal trends (Fig. S2). In particular, we found no directional trend in any of the community metrics according to these methodological factors. However, we note that shorter time series are often associated with faster changes (both negative and positive), as it is expected as longer time series aggregate changes over a longer time span. We further reference this new figure in the results and present it in the supplementary materials as Fig. S2. In response to comment #2 of the Reviewer 3, we also explored whether the characteristics of the time series were related to the ecological drivers we selected, and found that it was not the case (see new Fig. S5 below).

Figure S2: Covariations between temporal trends estimated at the site level and the characteristics of the time series. The year span is expressed in number of years, completeness in proportion of years over the entire study period and first year of sampling in year. The temporal trends were estimated with time as sole fixed predictor (See Methods, eq. (3)). The black lines display the regression lines fitted with a Generalized Additive Model (GAM) and the gray area the confidence interval.

source, provide a link to the Creative Commons license, and indicate if changes were made. In the cases where the authors are anonymous, such as is the case for the reports of anonymous peer reviewers, author attribution should be to 'Anonymous Referee' followed by a clear attribution to the source work. The images or other third party material in this file are included in the article's Creative Commons license, unless indicated otherwise in a credit line to the material. If material is not included in the article's Creative Commons license and your intended use is not permitted by statutory regulation or exceeds the permitted use, you will need to obtain permission directly from the copyright holder. To view a copy of this license, visit <http://creativecommons.org/licenses/by/4.0/>.

P6 - L 77-83: We estimated temporal trends with a hierarchical Bayesian modeling approach that accounts for spatial variation at both hydrographic river basin and site levels (i.e. by including random terms on the intercept and temporal trends; see Methods for a detailed description of the models), finding that communities have increased in both total abundance and species richness, but decreased in the proportional abundance of non-native species. We further found that the estimated temporal trends were not influenced by the characteristics of the time series, such as the temporal span, survey completeness and starting year (Fig. S2).

11. Page 21, lines 8–13: I am uncertain if 'present pressures' is the most suitable term to use it. It may be more accurately stated as the rate of change of the human footprint. Likewise, I don't want to undervalue the reliability of the human footprint index from Venter et al. (2016). However, I recommend that the authors provide a brief rationale for why they have specifically chosen to use the human footprint index from Venter et al. (2016) instead of considering other human footprint indices calculated in previous works (e.g., Ellis et al., 2020; Mu et al., 2022).

The reviewer is correct. We changed 'present pressures' to 'recent changes' where this expression was used (see examples below).

Page 17 L 300-302: Nonetheless, our analyses were conducted on the best available temporal riverine fish dataset at this time, and they provide evidence linking the multidimensionality of community changes to the interplay between past and recent environmental challenges as well as habitat context.

Page 17 L 305-306: In conclusion, our study uncovered complex but consistent effects of past and recent changes in anthropogenic pressures and stream network position on riverine fish communities.

We also provide a brief rationale for using the HFI developed by Venter et al. (2016) instead of other related indices. Specifically, we chose the Venter et al. (2016)'s index because its first year (i.e., 1993) better corresponded with the beginning of our time series as compared to the one from Mu et al. (2022) (i.e., 2000). It was especially important in the context of our study as we were interested in studying how ecological communities may respond to long-lasting disturbance effects as well as recent pressures. The Mu et al.'s index is certainly more appropriate when considering areas that developed most rapidly over the last decades such as Southern and Southeast Asia and Africa. However, as noted in Discussion, our time series are mainly located in Western Europe, North America and Southeast Australia, which are areas that industrialized far before the 2000s. Overall, a comparison between the two HFI indices for the year 2009 revealed a high congruence between the two (Spearman rho correlation = 0.81 in our dataset), despite the fact that the HFI values from Mu et al. (2022) tend to be higher than the ones from Venter et al. (2016) and display bounding effects at maximal values (HFI = 50) (see figure below). Regarding Ellis et al. (2020)'s index, we believe that its temporal extent (12k years) is more appropriate for studies that involve long-term temporal data such as in paleontology (e.g., palynology, paleoentomology). Lastly, we note that our choice was influenced by previous studies that demonstrated that Venter et al. (2016)'s HFI is an appropriate product to associate variation in biodiversity responses to human pressures (e.g., Di Marco et al. 2018; Di Marco et al. 2019), as well as by its integration (and thus availability) in the widely used RiverATLAS database (Linke et al. 2019). We added a sentence to justify the use of Venter et al. (2016) HFI.

P21 - L 399-402: To capture both the effects of the legacy of past anthropogenic pressures and of its recent changes, we considered the human footprint index computed in 1993 and 2009 (i.e. 16 years span). Specifically, the human footprint index of 1993 was used as a measure of past anthropogenic pressures at the beginning of the study period and the ratio between the human footprint of 2009 and 1993 as a measure of the recent changes in anthropogenic pressures (only 7% of the samplings took place before 1993, while 58% took place between 1993 and 2009 and 34% after 2009; Fig. S1). We chose the human footprint index developed by Venter et al.³⁰ because it goes farther back in time than other related human footprint indices (e.g. 1993 vs. 2000 in Mu et al.⁵¹). Nevertheless, both indices are highly correlated (Spearman's $\rho = 0.81$ for 2009) in overlapping years, suggesting that our results are robust to the choice of human footprint index.

12. Page 21, last two line: Indicate the positive and negative values from the longitudinal gradient.

source, provide a link to the Creative Commons license, and indicate if changes were made. In the cases where the authors are anonymous, such as is the case for the reports of anonymous peer reviewers, author attribution should be to 'Anonymous Referee' followed by a clear attribution to the source work. The images or other third party material in this file are included in the article's Creative Commons license, unless indicated otherwise in a credit line to the material. If material is not included in the article's Creative Commons license and your intended use is not permitted by statutory regulation or exceeds the permitted use, you will need to obtain permission directly from the copyright holder. To view a copy of this license, visit <http://creativecommons.org/licenses/by/4.0/>.

We thank the reviewer and we modified the sentence accordingly.

P22 - L 421-426: The first rotated component was positively related to average annual discharge, distance from source and Strahler order and captured 56% of the variance (Fig. S6), and was subsequently used as a composite variable describing the longitudinal stream position from upstream to downstream (from negative to positive values, respectively). The correlation coefficients indicated little covariations among the predictors (maximum Spearman's correlation of 0.09 found between past and recent pressures, Fig. S9).

13. Page 21, last line: Indicate type of correlation and "Spearman's correlation".

Revised (see comment #12).

14. Page 25, equation: Include number of the equation as it has been done in the previous ones.

Revised.

15. Page 26, lines 3–4: Would it be possible to include, within brackets, some comparisons between the temporal trends of non-native species and richness? This would make it easier for the reader to discern and understand any differences between them.

We now report the kurtosis of the distribution showing that non-native species temporal trends are much more heavy-tailed than species richness and refer to Fig.1.

P27 - L 534-539: We excluded the two variables describing the non-native species composition (proportion of non-native abundance and richness) because their predicted temporal trends at the site level displayed a very low variation compared to the other variables (kurtosis of the distributions: 10 vs 21 and 8 vs 15 for total vs non-native abundance and species richness, respectively; Fig. 1), which in turn exerted a disproportionate constraint on the analysis.

16. Figure 1. It may be better to increase the size of the dots in each barplot as I barely see them. Please, also consider including precise numerical values just besides the dots (% of change and CI). This could enable readers to effortlessly establish mental connections with the information given in the main text.

Thank you for the suggestion. We modified the figure accordingly.

17. Figure 2. I am unsure if panel b is necessary in this case given the low explained information provided by the third and fourth PCA axes. I suggest removing it or moving to the Appendix. By saying this, I understand that the cluster assignment for the distinct types of community trajectories was applied to only the first PCA axes, right? Otherwise, there should be betterer clarified it in the M&M.

source, provide a link to the Creative Commons license, and indicate if changes were made. In the cases where the authors are anonymous, such as is the case for the reports of anonymous peer reviewers, author attribution should be to 'Anonymous Referee' followed by a clear attribution to the source work. The images or other third party material in this file are included in the article's Creative Commons license, unless indicated otherwise in a credit line to the material. If material is not included in the article's Creative Commons license and your intended use is not permitted by statutory regulation or exceeds the permitted use, you will need to obtain permission directly from the copyright holder. To view a copy of this license, visit <http://creativecommons.org/licenses/by/4.0/>.

To clarify, the cluster assignment was performed independently from the PCA, on the estimated temporal trends at the site level. We recognize that it was unclear in the manuscript, so we revised the text in the Methods and Results. In view of this clarification, we decided to retain the panel b because it illustrates well the distinction between sites that displayed either temporal changes in species richness or in total abundance.

P27 - L 539-543: In addition, we performed a clustering analysis on the temporal trends in the community metrics at the site level to identify distinct types of community trajectories using the trimmed k-means method, a robust clustering method because it avoids the identification of spurious clusters. The method consists of trimming the most outlying data in the multidimensional space, the number of dimensions being the number of community metrics.

P9 - L 118-120: Using a k-mean trimmed clustering method on the temporal trends in the community metrics at the site level (see Methods), we further detected six distinct types of community trajectories (Fig. 2c).

References

- Ellis, E.C., Beusen, A.H.W., Goldewijk, K.K., 2020. Anthropogenic Biomes: 10,000 BCE to 2015 CE. *Land* 9, 129. <https://doi.org/10.3390/land9050129>
- Mu, H., Li, X., Wen, Y., Huang, J., Du, P., Su, W., Miao, S., Geng, M., 2022. A global record of annual terrestrial Human Footprint dataset from 2000 to 2018. *Sci Data* 9, 176. <https://doi.org/10.1038/s41597-022-01284-8>
- Strayer, D.L., Eviner, V.T., Jeschke, J.M., Pace, M.L., 2006. Understanding the long-term effects of species invasions. *Trends in Ecology & Evolution* 21, 645–651. <https://doi.org/10.1016/j.tree.2006.07.007>
- Venter, O., Sanderson, E.W., Magrath, A., Allan, J.R., Beher, J., Jones, K.R., Possingham, H.P., Laurance, W.F., Wood, P., Fekete, B.M., Levy, M.A., Watson, J.E.M., 2016. Global terrestrial Human Footprint maps for 1993 and 2009. *Sci Data* 3, 160067. <https://doi.org/10.1038/sdata.2016.67>

Signed: Ignasi Arranz

Reviewer 2

This study presented extremely interesting research on how past and recent human pressure has shaped fish biodiversity (richness, abundance, and composition) over time. A massive database is employed across a large spatial scale, making the results remarkably consistent. I find the results of this study of broad interest, especially to aquatic ecologists and fisheries managers. I found no major flaws in the structure and analysis of the manuscript. Although I do not consider the conclusion to be extremely new, they add very important information for understanding fish biodiversity changes and how human pressures contribute to such changes.

I have some major doubts and suggestions that I believe will improve the clarity of the study.

We thank the reviewer for their positive endorsement of our paper, and for providing such an in-depth review that has helped improve the clarity.

1. While I understand that the journal has word limitations, I think the authors could, in a few sentences in the last paragraph of the introduction, explain what was considered human pressure (in this case, human footprint). Particularly, what is the Human Footprint? and which metrics of human pressure layers this index considers? Still, would it be possible to draw the human footprint map: past (1997) and recent (2009), and make these two maps available, including the surveyed sites in the supplemental material? The HFP map is easily drawn with the coordinates of the sampling sites. I believe that a map of the HFP in 1993 and 2009 could improve the reader's visualization of the change in HFP over time.

Yes, we agree with the reviewer and added a sentence in the Introduction to provide more context regarding the metrics of anthropogenic pressures (see below). We also added the HFI of 1993 and its recent change (between 1993 and 2009) to the Supplementary Map to improve the presentation.

P4 - L 66-70: We finally assessed how fish community changes could be traced back to the spatio-temporal changes in anthropogenic pressures and longitudinal stream position. Anthropogenic pressures were quantified with the human footprint index, which includes an array of pressures such as population density, land-use and human-built infrastructure³⁰ (See Methods) and has been previously related to species extinction and invasion risks^{31,32}.

2. I have a concern/doubt about the structure of Bayesian models, especially the interaction between human footprint and time. Overall, human footprint and time are not independent; that is, the human footprint scale with time <https://doi.org/10.1038/s41597-022-01284-8>. It is also confusing how the interaction between human footprint vs time was considered if there are only two temporal measures of human footprint (1993 and 2009). For example, how was Time interacted with Past pressures if there was only Human footprint measurement in 1993? Time seems to be a category with only 1 level for the human footprint, how can we interpret

such an interaction? How could the HFP interact with Time if there is no temporal replicability of the HFP? If the authors really want to test the temporal effect of HFP, it might be necessary to use annual HFP values. There are annual HFP measurements from 2000 to 2018 (see: <https://doi.org/10.1038/s41597-022-01284-8>). If the intention is just to compare the effect of past and present HFP on biodiversity, perhaps the HFP vs. Time interaction is unnecessary since it is just comparing the slope of past and present HFP. I really don't know if the interaction between time and human footprint is necessary, since this was not discussed in the paper. In any case, a little more explanation of the interaction between HFP vs. Time should be provided.

The reviewer has offered an excellent comment that additional explanation is needed to clarify the choice of the interaction between time and HFI in the models. To provide some broader context, the majority of published studies have investigated the effect of land-use on spatial variation in biodiversity, specifically using a space-for-time substitution approach that does not explicitly consider how anthropogenic pressures affect temporal trends (e.g., Newbold et al 2015, Outhwaite et al 2022). By contrast, we were specifically interested by these temporal effects and used the interaction between Time and Past pressures to test whether the temporal trends in community composition (Time effect) was modulated by the HFI value in 1993 (where the HFI of 1993 was used as a proxy of past land use changes). Time was not a categorical variable but a discrete numerical variable, expressed in number of years since the first year of sampling at a given site. HFI was a continuous numerical variable. A positive/negative interaction value thus indicates that temporal trends in community composition increased/decreased when HFP values in 1993 were higher (indicating a higher past degree of anthropogenic pressures) (Fig. 3a).

To capture the effects of recent changes in anthropogenic pressures, we included temporal changes in HFI between 1993 and 2009 from Venter et al. (2016). Additional rationale is provided in our response to comment #11 of Reviewer 1. Similarly to the effects of past anthropogenic pressures, we included an interaction with Time to test whether the temporal trends in community composition increased/decreased when differences in HFI values between 1993 and 2009 were positive/negative (i.e., when the sites experienced an increase or decrease in anthropogenic pressures in recent years) (Fig. 3a).

We note that in the last section of the Results, we also explored whether spatial variation in community composition relates to spatial variation in the degree of past anthropogenic pressures (HFI of 1993) and recent changes in anthropogenic pressures (changes in HFI between 1993 and 2009), independently from Time (Fig. 3b). As such, our objective was not to simply compare the effects of past (1993) and recent (2009) HFI on biodiversity, but to compare the effects of past and recent changes in HFI on biodiversity temporal trends and spatial variability. We have added a sentence to clarify why the interaction between time and anthropogenic drivers was necessary.

P23 - L 447-456: We included two-way interactions between time and the ecological drivers (Equation) to test how longitudinal stream position and anthropogenic pressures affect the temporal trends in community metrics. For instance, a positive interaction between time and past anthropogenic pressures would indicate that faster changes were happening in historically degraded areas. We further included the three-way interactions between time and the pairs of other ecological drivers (Equation) to examine the presence of synergistic or antagonistic effects between anthropogenic pressures and longitudinal stream position on the temporal trends in community metrics (i.e., the results in Fig. 3a). For instance, a positive interaction between time, recent changes in anthropogenic pressures and longitudinal stream position would indicate that the effects of recent changes in anthropogenic pressures on temporal trends were more important in downstream areas.

3. It appears that the authors have switched the colors of the proportion of non-native richness and abundance in Fig. 3 and in the text. For example, in the text, the authors state that the effect of past anthropogenic pressures on the proportion of non-native richness is displayed in yellow, and on the proportion of non-native abundance is displayed in orange (page 10: second paragraph). However, in Fig. 3 proportion of non-native richness is displayed in orange while the proportion of non-native abundance is displayed in yellow. Please, fix this. The slope of the interaction between Time × Stream long. position × past pressures on the proportion of non-native abundance do not overlap with zero (Fig. 3a). Doesn't this indicate a strong effect of the past anthropogenic pressure on the non-native abundance in most downstream areas (following the interpretation of 80% = weak, 90% = moderate, and 95% = strong)? If yes, consider changing the "moderate evidence" to "strong evidence" in the text.

We apologize for this oversight. There was indeed an inversion of color in the text. We modified "orange" to "yellow" in the text. The reviewer is also right about the interaction, we thus changed "moderate" to "strong" effect.

4. Although remarkably consistent, this study is correlational. I missed some consideration of this in the discussion or elsewhere in the manuscript.

We agree with the reviewer and added a sentence about this aspect of the study (see response to Reviewer 3 #5).

5. Page 9: What do you mean by "higher degree of past anthropogenic pressures"? Perhaps it is better: "increase in past anthropogenic pressures".

We are referring to the sites that displayed a higher HFI in 1993 (that we used as a proxy of degree of past anthropogenic pressures). We hope that our response to comment #2 clarified this point.

6. Page 15 (second paragraph). It is stated that: "The increase in local species richness over time could thus result from the introduction of non-native species". Doesn't this

statement contradict the results of Fig. 1, in which the richness of non-native species does not change and the abundance of invasive species decreases over time? Fig. 1 (c and d) seems to show no increase in the richness and abundance of non-native fish species over time. Perhaps you mean: "The increase in local species richness in highly degraded rivers over time could thus result from the introduction of non-native species".

Yes, the reviewer is correct. We modified the sentence accordingly.

P15 - L 245-246: The increase in local species richness over time in degraded rivers could thus result from the introduction of non-native species from ongoing spatial homogenization.

Reviewer 3

This manuscript uses time series of river fish assemblages to assess how various metrics of biodiversity differ in space and over time and in relation to past and recent human impact. The paper finds change in community composition and increases in both abundance and richness over the time series assessed. They also assess how the proportion of non-native species richness and abundance has changed over time, which could explain some of the positive trends in abundance and richness. This was a valuable addition as positive trends in freshwater biodiversity are often contested, going beyond the usual metrics to assess change in more detail.

I found the content of the manuscript interesting, I thought the interactive map was a nice touch, and it was very impressive that the document was written in R Markdown.

There are however a number of things that I think need to be addressed before the work can be published within this journal.

We thank the reviewer for their positive comments on our manuscript. We addressed all the comments raised below.

1. Limited description of the data and methods in the main text.

I found it very challenging to fully understand what the data included and how the analyses were carried out from reading the main text of this manuscript. In those journals where the methods are after the rest of the main text, I think it is really important to be able to get a good understanding of what the data are, and briefly, how they were assessed.

It is really unclear, for example, what the time series data represent, where they are from, how they were collected and the general coverage from the information given in the introduction and results. The final paragraph of the introduction could include much more information on the data, data coverage, and analysis as well as linking to the specific analyses of interest.

Things that could be made more clear in particular – the last 3 decades is used consistently to describe the coverage, but the data used when you get to the method extends further back than this in some cases; across three biogeographic realms – descriptions of the data present three main regions, but then the map presents data from elsewhere.

We agree with the reviewer that details regarding the data could have been further described in the manuscript. As requested by the reviewer, we added additional information regarding the time series in the text as well as a figure in the Supplementary Material that better describes the temporal coverage of the data. We have only six sites out of 4,476 total that are located outside of the three main biogeographic realms.

P3 - L 56-62: This study investigates the spatio-temporal changes of riverine fish communities in response to human pressures from local to continental extents. To do so, we leveraged a compilation of 4,476 riverine fish community time series that had been repeatedly sampled over 1957 to 2019 using variable durations and frequencies (93% of the samplings between 1993 and 2019 with a minimum of 5 years of sampling, Fig. S1), mainly using electrofishing (98% of the samplings). The sites are located in various river basins, mainly across the Palearctic, Nearctic and Australasia realms (99.9% of the sites).

2. Sensitivity tests and other checks

The authors describe their methods used in relation to other studies that have followed a similar analytical framework, often stating tests that these other papers cover, however there very few explorations or tests carried out on the data used here to provide reassurance for the presented results. Some questions that came to my mind when reading included: Does the length of the time series impact the results (i.e. are longer time series more likely to be increasing? Since they have had longer to recover from past impact. Do time series that were collected in more recent years or that start further back have an influence on the overall results? Is there any bias in terms of geographic region? Does each region cover similar time frames, lengths of time series, range of predictors? Is there an influence of start year on clustering? E.g. Are earlier/later time series associated with certain types of change? It would be reassuring to see the authors address some points to explore the potential influence of certain subsets of the data, particularly give the criticisms that many of the previous studies that this one cites within the methods have faced.

Our study included a robust selection procedure of time series where we retained just 32% of the time series available over a three-decade time period from the RivFishTIME database (Comte et al. 2021) and additional datasets (Table S2). However, in response to this concern, we have now conducted additional analyses and present additional figures showing that the starting year, the completeness, and time span of the time series have no overall effects on the estimation of the temporal trends at the site scale (details regarding these new analyses are provided in our response to comment #10 from Reviewer 1). In response to this comment, we also explored whether the characteristics of the time series were related to the ecological drivers we selected, and found that it was not the case (see new Fig. S5 below).

P 9 - L 132-136: We detected complex synergies between the legacy of past anthropogenic pressures and the effects of recent anthropogenic pressures on community temporal trends, by considering additional predictors associated with the human footprint index and longitudinal stream position (Fig. 3a; see model predictions in Fig. S6). In addition, we found that these additional predictors were not related to the characteristics of the time series (Fig. S5).

source, provide a link to the Creative Commons license, and indicate if changes were made. In the cases where the authors are anonymous, such as is the case for the reports of anonymous peer reviewers, author attribution should be to 'Anonymous Referee' followed by a clear attribution to the source work. The images or other third party material in this file are included in the article's Creative Commons license, unless indicated otherwise in a credit line to the material. If material is not included in the article's Creative Commons license and your intended use is not permitted by statutory regulation or exceeds the permitted use, you will need to obtain permission directly from the copyright holder. To view a copy of this license, visit <http://creativecommons.org/licenses/by/4.0/>.

Figure S5: Covariations between ecological drivers used in the hierarchical Bayesian model (longitudinal stream position, past and recent changes in anthropogenic pressures) and time series characteristics (time span in number of years and beginning of the time series in year). The blue lines display the regression lines fitted with a Generalized Additive Model (GAM) and the gray area the confidence interval. The ecological drivers were largely unrelated to the characteristics of the time series (N = 4,476 sites).

Following the reviewer comments, we further added a figure detailing the temporal characteristics of the sites according to the biogeographical realms. We found no evidence for strong biases, except for the time series from Australia that all begin in 2003. This figure has been integrated into the Supplementary Material (Fig. S1e). However, we note that by using hierarchical Bayesian models that account for the spatial variation across biogeographic basins and sites on the intercept and temporal trends (slope), the models control for potential geographical unbalance among the time series.

Figure S1: Distribution of (a) year of all sampling events (b) first year of the time series, (c) time span, (d) completeness of the time series, (e) time series of yearly sampling events. Colours

source, provide a link to the Creative Commons license, and indicate if changes were made. In the cases where the authors are anonymous, such as is the case for the reports of anonymous peer reviewers, author attribution should be to 'Anonymous Referee' followed by a clear attribution to the source work. The images or other third party material in this file are included in the article's Creative Commons license, unless indicated otherwise in a credit line to the material. If material is not included in the article's Creative Commons license and your intended use is not permitted by statutory regulation or exceeds the permitted use, you will need to obtain permission directly from the copyright holder. To view a copy of this license, visit <http://creativecommons.org/licenses/by/4.0/>.

correspond to the biogeographic realms. In (a), (b), and (e), the lines display the year 1993 and 2009. The years 1993 and 2009 correspond to the years of the human footprint index assessments, which were used to quantify past and recent changes in anthropogenic pressures (See Methods).

Minor comments:

There were no line numbers so apologies if it is not clear where these comments relate to.

Abstract:

3. The description of the data is very vague. "across three biogeographic realms", it would be good to list them here. Ideally, this should also give some idea of the timescale over which trends are being assessed.

We thank the reviewer, we added more details in the abstract.

Abstract: Understanding how and why local communities change is a pressing task for conservation, especially in freshwater systems. It remains challenging because of the complexity of biodiversity changes, driven by the spatio-temporal heterogeneity of human pressures. Using a compilation of riverine fish community timeseries (93% between 1993-2019) across the Palearctic, Nearctic and Australasia realms, we assessed how past and recent anthropogenic pressures drive community changes across both space and time. We found evidence of rapid changes in community composition of 30% per decade characterised by significant changes in the dominant species, together with a 13% increase in total abundance per decade, and a 7% increase in species richness per decade. The spatial heterogeneity in these trends could be traced back to the strength and timing of anthropogenic pressures and was mediated by non-native species introductions. Specifically, we demonstrate that the negative effects of anthropogenic pressures on species richness and total abundance were compensated over time by the establishment of non-native species, a pattern consistent with previously reported biotic homogenization at the global scale. Overall, our study suggests that accounting for the complexity of community changes and its drivers is a crucial step to reach global conservation goals.

Introduction:

4. Rather than "the last 3 decades", can you be specific about data coverage. Repeatedly sampled over the last 3 decades – sounds like all time series have the same start/end and duration. Alternatively; "repeatedly sampled for variable durations and frequencies between 19XX and 20XX."

We thank the reviewer, we incorporated this suggestion in the response to point #1.

5. Expect to see something about how human pressures have changed over the time period assessed, perhaps specifically in relation to where the data are from. Since most data are from developed northern hemisphere, are most regions likely to be improving in terms of water quality at least?

We added the map of past anthropogenic pressures and recent changes to the Supplementary Map (see response to Reviewer #2 comment #2).

Based on the recent changes in the human footprint index, it is apparent that anthropogenic pressures have decreased in our dataset by 0.9 times on average between 1993 and 2009, while it has increased by 1.5 times when all the river segments of the world are considered (data from RiverAtlas). This most certainly reflects the fact that most of our sites are located in the Northern hemisphere (see point in Discussion L285-294).

The human footprint index is an indirect measure combining an array of stressors (associated with land-use, transportation infrastructure and population density), which are known to be relevant to predict the overall ecological integrity of rivers (Lessmann et al 2019). However, even if water quality close to urban areas has improved due to the installation of wastewater treatment plants over the last decades, agricultural practices are still vastly contributing to eutrophication and pesticides diffusion in the Northern hemisphere (EEA report 2021). In addition, in Europe, aquatic biodiversity is still suffering from wetland drainage, damming and river channelisation, which collectively make rivers extremely vulnerable to extreme events such as floods and droughts (EEA report 2021). Past pollution can also drive long remanence into rivers notably by their accumulation into the sediment (Rasmussen et al 2015). As such, even if the human footprint index points at a decrease in anthropogenic pressures across our dataset, some sites might still experience some ongoing degradations, with consequences for the fish communities. However, capturing the full array of threats affecting freshwater ecosystems and their interactive effects remains challenging (Craig et al 2017, Simmons et al 2021). So, we added a discussion point about the importance to consider more precise threat estimates at the local scale, but recognising the difficulty to account for multiple interactive effects among stressors.

P17 - L 297-300: Our study remains essentially correlative. Although aggregated anthropogenic threat indices have been shown to be useful to estimate the ecological integrity of freshwater ecosystems⁵³, they do not replace the use of more targeted threat indices related to water quality and ecosystem functioning^{54,55}.

6. Not clear to me what is meant by longitudinal stream position. There are also a number of terms that seem to be specialist for the freshwater realm throughout the that would benefit from explanation (reach, Stahler order)

We incorporated explanation in the Introduction (See answer to comments of the Reviewer 1 #3) and the Methods as well.

Page 21 L391-393: *We quantified the degree of anthropogenic pressures using the human footprint index, extracted from the RiverAtlas database at the reach scale^{52,64} (stream segment length average of 450 metres).*

Page 22 L413-416: *To capture this longitudinal gradient, we first described stream characteristics at each site by the altitude (m), slope (deg.), average annual discharge (m³.s⁻¹), distance from source (km), and Strahler order (i.e. downstream position based on stream/tributary hierarchy) that we extracted from the RiverAtlas database at the stream segment scale^{52,64}.*

7. No information on how the data were analysed. Suggest something starting like “ We used Bayesian hierarchical models to assess change in fish abundance, richness... as a function of time and of ...”.

We thank the reviewer for the suggestion. We added the mention of the type of model used to analyze the data.

P3 - L 62-67: *We used Bayesian hierarchical models to assess temporal changes in total abundance, species richness, and community composition across local communities, including in the share of non-native species. We next characterised the typology of community temporal trends by examining the covariations among different community metrics, and identifying trajectories of community change across spatial scales. We finally assessed how fish community changes could be traced back to the spatio-temporal changes in anthropogenic pressures and longitudinal stream position.*

Results:

Community temporal trends

8. In the first paragraph, the first set of results are repeated, the first instance can be removed. “strong evidence” and switching between 80/90/95% CIs. Since this is not general practise and not often seen in papers, I think it might be best either to explain this approach in the main text, not just in the methods, or to stick to use of one set of CI ranges. In the second paragraph of the results, no values are given but just “(strong evidence)” supplied in brackets. If the methods have not been read first, I think this could be potentially confusing to a reader. Suggest removing the (strong evidence) either way.

We agree with the suggestion of the reviewer and we added an explanation in the first paragraph of the result. We also removed the (strong evidence) supplied in brackets in the second paragraph.

P6 - L 77-86: *We estimated temporal trends with a hierarchical Bayesian modeling approach that accounts for spatial variation at both hydrographic river basin and site levels (i.e. by*

including random terms on the intercept and temporal trends; see Methods for a detailed description of the models), finding that communities have increased in both total abundance and species richness, but decreased in the proportional abundance of non-native species. We further found that the estimated temporal trends were not influenced by the characteristics of the time series, such as the temporal span, survey completeness and starting year (Fig. S2). From this model, we considered weak, medium and strong evidence for an effect when its confidence interval at respectively 80, 90, and 95% did not overlap 0^{33,34}. We found strong evidence for an average increase in total community abundance (average Credible Interval 95%: 13.2% [2.9%,23.8%] per decade, Fig. 1a) and in species richness [...]

9. Paragraph 3 – time as a random slope, but Fig 1 legend says time as the only predictor. It is not clear from the current explanation what the models actually represent without having to go to the methods. Description so far in the main text indicates models include time with random effects of basin and site only.

We agree with the reviewer and we added the precision that the random effects of basin and site were added on both the intercept and the slope (See response to your comment #8). We further added to the legend of Fig. 1 that time is the sole fixed predictor.

Fig 1 legend: Distribution of community temporal trends across sites: (a) total abundance, (b) species richness, (c) proportion of non-native total abundance, (d) proportion of non-native species richness, (e) Simpson temporal dissimilarity, and (f) Jaccard turnover. Temporal trends per decade were estimated from a hierarchical Bayesian model including time as sole fixed predictor and using a random slope to estimate temporal trends at each site (See Methods).

Typology of community temporal trends

10. Starting description could be altered to briefly describe the method and how/why it was used before going into the results. At the moment, this section doesn't flow well in terms of what was done and why.

We agree with the reviewer and we added a sentence to describe the goal of this analysis and the data used to do it.

P9 - L 111-115: We further assessed covariations among the temporal trends of different community metrics to identify potential "types" of community temporal trajectories using the temporal trends at the site level estimated from the hierarchical Bayesian model. There was a moderate level of association among the different community trajectory metrics; the first two axes of the principal component analysis explained 69% of the total variability among fish communities (Fig. 2a-b).

11. Figure 2 legend, description of cluster names might be helpful here, especially for medium change which is not very clear without reading the methods.

We agree with the reviewer. We made more explicit the link between the cluster names and the characteristics of the temporal changes in panel c and added a description of the “medium change” cluster.

Fig. 2 legend: Figure 2: Covariation among the community temporal trends and characterization of community trajectories. (a, b) PCA biplot of the community temporal trends and their cluster assignment where the sites are colored according to their cluster assignment. (c) Boxplots displaying the distribution of the temporal trends by cluster. The boxplot limits display 1.5 interquartile range. (d) Cluster frequencies across the three main biogeographic realms. The ellipses in (a-b) display the 95% intervals around the clusters assuming a Student's t distribution. The clusters were named according to the most noticeable characteristic of changes across all the biodiversity metrics (panel c). In particular, 'medium change' cluster was associated with sites presenting moderate changes along all the biodiversity metrics considered. Sites not assigned to a cluster because of affiliation uncertainty (N = 641, 14%) are displayed in Fig. S4. N = 46,932 sampling events across 4,476 sites.

Drivers of community temporal trends

12. What is Strahler order, needs explanation for wide audience journal.

We agree and we added an explanation, see our response to point #6

13. Non-native species abundance referred to as orange in the figure but it is actually yellow.

We apologize for this oversight. We modified the color reference in the text (See response to Reviewer 2 #4).

14. Figure S4 – repeat of non-native abundance. Think one should be richness. Also line and ribbon plot might be easier to visualise than 3 lines here.

We thank the reviewer. We corrected the legend and improved the visualisation here (Fig. S4 became S6).

Fig S6 legend: Predictions of community metric changes over time according to past human pressures and recent changes in human pressures using a hierarchical Bayesian model. Total abundance is expressed in count, species richness in number of species (raw), while non-native abundance, non-native richness, dissimilarity and turnover are expressed in proportion.

Drivers of community variation across space

15. It is not clear what the model is that determines these results. Please make it clear what variables are being assessed in each instance.

We agree with the reviewer that the origin of the numbers was not clear. We added that the numbers come from the prediction of the model at $t = 0$ (Fig. S4). We also added a paragraph in the Methods to clarify how we controlled for the values of the other predictors.

Page 13 L190-197: Spatial variation in community structure was strongly associated with past and recent anthropogenic pressures and with longitudinal stream position (i.e. single model effects independent of time, Fig. 3b). Using baseline model prediction (i.e. at $t = 0$, Fig. S6, see Methods), we found that a higher degree of past anthropogenic pressures was associated with lower total abundance (strong evidence; Fig. 3b, blue), with the most 'degraded' sites (i.e. with a human footprint index = 45.6) displaying a total abundance 30% lower than the most 'intact' sites (i.e. with a human footprint index = 2.5). By contrast, a higher degree of past anthropogenic pressures was associated with higher species richness (strong evidence; Fig. 3b, green), with the most degraded sites displaying 64% more species than the most intact sites.

Methods L465-468: Finally, we derived the comparison in community metrics across space according to the ecological drivers by using the predictions of the model at the baseline ($t = 0$). The predictions controlled for the values of other predictors (such as longitudinal stream position and past anthropogenic pressures) by setting them at their median values.

16. Do these models account for year of sampling at all?

No, the models do not account for the year of sampling at all. In response to the comment #10 of Reviewer 1, we showed that the characteristics of the time series such as the first year of sampling, completeness and time span have no overall effect on the estimated temporal trends.

17. Most upstream rather than upstream-most.

We changed the text.

Discussion

18. Why do you think the results here differ from the global trends?

We think that the results could differ because the human footprint index has decreased by 4% on average across our sites while it increased by 25% globally.

P17 L290-294: We recognise that recent increases in anthropogenic pressures are most prevalent in tropical biodiversity hotspots, while our study has a spatial coverage limited to historically industrialised countries and in mostly temperate biomes. Noticeably, anthropogenic pressures decreased by 4% on average (based on differences in the human footprint index between 1993 and 2009) across the sites included in our study while it increased by 25% across the rivers globally⁵².

19. What about the time point chosen for the past? What impact could this have on results?

Especially considering most of the data come from regions where pressures have been reducing in freshwater systems compared to further back in the past. There are certainly

some points around baseline and reference points that could be discussed and potentially explored with some additional tests.

We agree that the point of reference is an important matter in time but also in space, according to the timing of the pressures. We added a point about the implication of our results compared to countries that experienced more recent increase in human pressures (see our answer to comment #18). We further did additional tests showing that the beginning year of our time series does not affect the estimated temporal trends (see our answer to Reviewer 1 comment #10).

Methods

20. The description of the data suggest data from certain regions only, then the interactive map shows other regions are included. Please be clear about where the data are from. I think it would be useful to have a map of the sites in the main text.

We added in Introduction and in the Abstract that the three main regions gather 99% of the sites, please see our response to comments #1 and #3. We are inclined to keep only the map in Supplementary Material but note that we added layers of anthropogenic pressures (past and recent) to improve the visualization. However, we are willing to add a map of the sites in the main text if deemed more appropriate.

21. What were the data sampling, were they sampling everything, certain species? Very little information to help understand the data and the potential biases. The quantiles are presented for the data, I think ranges would be more informative.

We agree with the reviewer that the description was too succinct. We added information about the methods of sampling, as well as the temporal span of the data in the last paragraph of the Introduction in response to the comment #1 of Reviewer 3. We now present both the interquartile and the range.

22. The description of the hierarchical nature of the data, sites within basins (?) could be improved.

We agree and added information about the hydrographic basins.

P18, L334-336: The sites were mostly located in the Palearctic (75%), Nearctic (20%) and Australasia (5%), and distributed across 307 hydrographic basins. Four countries gathered 85% of the sites, namely Great Britain (29%), France (21%), Sweden (18%), and the USA (18%, Table S3).

23. Dissimilarity – first year as reference, but first year varies between time series. Could this influence the results?

We showed in response to the comment #10 of Reviewer 1 that the starting year of the time series showed no significant association with the estimated dissimilarity trends, so we are confident that the starting year does not affect our results.

24. The methods section (and linked to the final paragraph of the introduction actually) would benefit from clearer relation of questions/areas being explored to the models being run.

We agree. We clarified in the methods that we built two different models and which one was used for each analysis.

P23, L443-447: To assess the drivers of community change, we then built a second model incorporating additional covariates (eq. (4)): the degree of past anthropogenic pressures measured by the human footprint index of 1993, the recent changes in anthropogenic pressures measured by the ratio between the human footprint index of 2009 and 1993 (X_k), and the longitudinal stream position estimated by the rotated PCA axis.

P25, L530-533: To estimate the covariations among multiple dimensions of community change, we performed a PCA on the temporal trends in the community metrics estimated at each site with the models (eq. (3)) having time as a sole fixed predictor (i.e. using the predictions in percentage per decade using the Best Linear Unbiased Prediction method).

Decision Letter, first revision:

24th October 2023

Dear Dr. Danet,

[REDACTED]

Thank you for submitting your revised manuscript "Past and recent anthropogenic pressures drive rapid changes in riverine fish communities" (NATECOLEVOL-23051182A). It has now been seen again by the original reviewers and their comments are below. The reviewers find that the paper has improved in revision, and therefore we'll be happy in principle to publish it in Nature Ecology & Evolution, pending minor revisions to satisfy the reviewers' final requests and to comply with our editorial and formatting guidelines.

Please email us a copy of the file in an editable format (Microsoft Word or LaTeX)-- we can not proceed with PDFs at this stage.

[REDACTED]

Reviewer #1 (Remarks to the Author):

I have completed a second review of the manuscript titled "Past and recent anthropogenic pressures drive rapid changes in riverine fish communities" for its publication in the journal Nature Ecology and Evolution. The authors did an excellent job in addressing all my concerns and I believe that the manuscript does not present any major issues. Please, consider these last minor points, but there is no need, at least to me, for another round of revisions.

L22: add "mainly" before "mediated".

L48: I am not sure if "trajectories" is the most suitable word in this context. Consider rephrasing it as "Temporal changes in community composition".

L102: add "spatial" after "same".

37Reviewer #2 (Remarks to the Author):

The authors have responded commendably to my previous comments. Therefore, I have no further consideration. I believe this manuscript will be an important contribution to ecology and I can't wait to see it published.

Reviewer #3 (Remarks to the Author):

The authors have done a fantastic job addressing the comments of the reviewers. All of my comments have been addressed, so thank you. The paper reads extremely well and I find the main text much easier to follow and to be more descriptive of what was conducted than in the previous version.

I do not have any further comments to make and recommend this work for publication.

Our ref: NATECOLEVOL-23051182A

1st November 2023

Dear Dr. Danet,

Thank you for your patience as we've prepared the guidelines for final submission of your Nature Ecology & Evolution manuscript, "Past and recent anthropogenic pressures drive rapid changes in riverine fish communities" (NATECOLEVOL-23051182A). Please carefully follow the step-by-step instructions provided in the attached file, and add a response in each row of the table to indicate the changes that you have made. Please also check and comment on any additional marked-up edits we have proposed within the text. Ensuring that each point is addressed will help to ensure that your revised manuscript can be swiftly handed over to our production team.

****We would like to start working on your revised paper, with all of the requested files and forms, as soon as possible (preferably within two weeks). Please get in contact with us immediately if you anticipate it taking more than two weeks to submit these revised files.****

When you upload your final materials, please include a point-by-point response to any remaining

38reviewer comments.

In recognition of the time and expertise our reviewers provide to Nature Ecology & Evolution's editorial process, we would like to formally acknowledge their contribution to the external peer review of your manuscript entitled "Past and recent anthropogenic pressures drive rapid changes in riverine fish communities". For those reviewers who give their assent, we will be publishing their names alongside the published article.

Nature Ecology & Evolution offers a Transparent Peer Review option for new original research manuscripts submitted after December 1st, 2019. As part of this initiative, we encourage our authors to support increased transparency into the peer review process by agreeing to have the reviewer comments, author rebuttal letters, and editorial decision letters published as a Supplementary item. When you submit your final files please clearly state in your cover letter whether or not you would like to participate in this initiative. Please note that failure to state your preference will result in delays in accepting your manuscript for publication.

Cover suggestions

We welcome submissions of artwork for consideration for our cover. For more information, please see our https://www.nature.com/documents/Nature_covers_author_guide.pdf guide for cover artwork.

Nature Ecology & Evolution has now transitioned to a unified Rights Collection system which will allow our Author Services team to quickly and easily collect the rights and permissions required to publish your work. Approximately 10 days after your paper is formally accepted, you will receive an email in providing you with a link to complete the grant of rights. If your paper is eligible for Open Access, our Author Services team will also be in touch regarding any additional information that may be required to arrange payment for your article.

Please note that *Nature Ecology & Evolution* is a Transformative Journal (TJ). Authors may publish their research with us through the traditional subscription access route or make their paper immediately open access through payment of an article-processing charge (APC). Authors will not be required to make a final decision about access to their article until it has been accepted.

39[href="https://www.springernature.com/gp/open-research/transformative-journals">](https://www.springernature.com/gp/open-research/transformative-journals) Find out more about Transformative Journals

Authors may need to take specific actions to achieve compliance with funder and institutional open access mandates. If your research is supported by a funder that requires immediate open access (e.g. according to Plan S principles) then you should select the gold OA route, and we will direct you to the compliant route where possible. For authors selecting the subscription publication route, the journal's standard licensing terms will need to be accepted, including https://www.nature.com/nature-portfolio/editorial-policies/self-archiving-and-license-to-publish. Those licensing terms will supersede any other terms that the author or any third party may assert apply to any version of the manuscript.

For information regarding our different publishing models please see our Transformative Journals page. If you have any questions about costs, Open Access requirements, or our legal forms, please contact ASJournals@springernature.com.

[REDACTED]

[REDACTED]

Reviewer #1:

Remarks to the Author:

I have completed a second review of the manuscript titled "Past and recent anthropogenic pressures drive rapid changes in riverine fish communities" for its publication in the journal Nature Ecology and Evolution. The authors did an excellent job in addressing all my concerns and I believe that the manuscript does not present any major issues. Please, consider these last minor points, but there is no need, at least to me, for another round of revisions.

L22: add "mainly" before "mediated".

L48: I am not sure if "trajectories" is the most suitable word in this context. Consider rephrasing it as "Temporal changes in community composition".

L102: add "spatial" after "same".

40Reviewer #2:

Remarks to the Author:

The authors have responded commendably to my previous comments. Therefore, I have no further consideration. I believe this manuscript will be an important contribution to ecology and I can't wait to see it published.

Reviewer #3:

Remarks to the Author:

The authors have done a fantastic job addressing the comments of the reviewers. All of my comments have been addressed, so thank you. The paper reads extremely well and I find the main text much easier to follow and to be more descriptive of what was conducted than in the previous version.

I do not have any further comments to make and recommend this work for publication.

Author Rebuttal, first revision

Response to reviewers

Legend:

- **Response**

Reviewer #1:

Remarks to the Author:

I have completed a second review of the manuscript titled "Past and recent anthropogenic pressures drive rapid changes in riverine fish communities" for its publication in the journal Nature Ecology and Evolution. The authors did an excellent job in addressing all my concerns and I believe that the manuscript does not present any major issues. Please, consider these last minor points, but there is no need, at least to me, for another round of revisions.

41Thank you very much for the positive feedback and the careful review.

L22: add “mainly” before “mediated”.

Fixed

L48: I am not sure if “trajectories” is the most suitable word in this context. Consider rephrasing it as “Temporal changes in community composition”.

Fixed

L102: add “spatial” after “same”.

Fixed

Reviewer #2:

Remarks to the Author:

The authors have responded commendably to my previous comments. Therefore, I have no further consideration. I believe this manuscript will be an important contribution to ecology and I can't wait to see it published.

Thank you very much for the positive and encouraging feedback.

Reviewer #3:

Remarks to the Author:

The authors have done a fantastic job addressing the comments of the reviewers. All of my comments have been addressed, so thank you. The paper reads extremely well and I find the main text much easier to follow and to be more descriptive of what was conducted than in the previous version.

I do not have any further comments to make and recommend this work for publication.

Thank you very much for the positive feedback.

Final Decision Letter:

13th November 2023

Dear Dr Danet,

I am delighted to tell you that your manuscript (NATECOLEVOL-23051182B) has been accepted for publication in Nature Ecology & Evolution.

We have now transitioned to a unified Rights Collection system which will allow our Author Services team to quickly and easily collect the rights and permissions required to publish your work. Once your paper is typeset, you will receive an email with a link to choose the appropriate publishing options for your paper and our Author Services team will be in touch regarding any additional information that may be required.

Once this step is complete, your proof will be made available through our e.proofing system via a link in a forthcoming communication. The system will show you an HTML version of the article that you can correct online. We may make minor changes to enhance the lucidity of the text, and if necessary to make it conform to the journal's style. We therefore ask that you examine the proofs most carefully to ensure that we have not inadvertently altered the sense of your text in any way.

****Please note that you will not receive your proofs until the publishing agreement has been received.****

Due to the quick turn-around of this content, it is important that you let us know if you will be unreachable for the next 2-3 weeks after acceptance, in which case we ask that you kindly send us the contact information of someone who will be able to check the proofs and handle any last-minute problems. Please address any correspondence about your proofs to our Production team at rjsproduction@springernature.com.

Please note that we might decide to distribute a press release to news organizations worldwide. This is usually done about ten days before the paper is published and it could include details of your work. We are more than happy for your institution or funding agency to prepare its own press release, but it must mention the embargo date and Nature Ecology & Evolution. Please contact us for more details.

Acceptance is conditional on the manuscript's not being published elsewhere, and on there being no announcement of this work to the newspapers, magazines, radio or television before the publication date. Nature Ecology & Evolution, however, does allow the registered journalists who receive our press

43release to have copies of papers a week before publication under strict embargo conditions, solely for the purpose of publicizing the work in the media. We permit these journalists to show papers to independent specialists a few days in advance of publication, again under embargo conditions, solely for the purpose of commenting on the work described. These restrictions are not intended to deter you from presenting your data at academic meetings and conferences, but any inquiries from the media about the papers not yet scheduled for publication should be referred to us.

[REDACTED]